# Anthropogenic emission controls reduce summertime ozone-temperature sensitivity in the United States

Shuai Li[1,2,3], Haolin Wang[1,2,3], Xiao Lu[1,2,3],*

[1] School of Atmospheric Sciences, Sun Yat-sen University, and Southern Marine Science and Engineering Guangdong Laboratory (Zhuhai), Zhuhai, Guangdong 519082, China

[2] Guangdong Provincial Observation and Research Station for Climate Environment and Air Quality Change in the Pearl River Estuary, Zhuhai, Guangdong 519082, China

[3] Key Laboratory of Tropical Atmosphere-Ocean System, Ministry of Education, Zhuhai, China, Zhuhai, Guangdong 519082, China

**Correspondence to** Xiao Lu (luxiao25@mail.sysu.edu.cn)

**Abstract**

The ozone-temperature sensitivity is widely used to infer the impact of future climate warming on ozone. However, trends in ozone-temperature sensitivity and possible drivers remained unclear. Here, we show that the observed summertime surface ozone-temperature sensitivity, defined as the slope of the best-fit line of daily anomaly in ozone versus maximum temperature ($m_{\Delta O3-\Delta Tmax}$), has decreased by 50% during 1990-2021 in the continental United States (CONUS), with a mean decreasing rate of -0.57 ppbv K$^{-1}$ per decade (p<0.01) across 608 monitoring sites. We conduct high-resolution GEOS-Chem simulations in 1995-2017 to interpret the $m_{\Delta O3-\Delta Tmax}$ trends and underlying mechanisms in the CONUS. The simulations identify the dominant role of anthropogenic nitrogen oxides (NO$_x$) emission reduction in the observed $m_{\Delta O3-\Delta Tmax}$ decrease. We find that approximately 76% of the simulated decline in $m_{\Delta O3-\Delta Tmax}$ can be attributed to the temperature-indirect effects arising from the shared collinearity of other meteorological effects (such as humidity, ventilation, and transport) on ozone. The remaining portion (24%) is mostly due to the temperature-direct effects, in particular four explicit temperature-dependent processes, including the biogenic volatile organic compounds (BVOCs) emissions, soil NO$_x$ emissions, dry deposition, and the thermal decomposition of peroxyacetyl nitrate (PAN). With reduced anthropogenic NO$_x$ emissions, the expected ozone enhancement from temperature-driven BVOCs emissions, dry deposition, and PAN decomposition decreases, contributing to the decline in $m_{\Delta O3-\Delta Tmax}$. However, soil NO$_x$ emissions increase $m_{\Delta O3-\Delta Tmax}$ with anthropogenic NO$_x$ emission reduction, indicating an increasing role of soil NO$_x$ emissions in shaping the ozone-temperature sensitivity. As indicated by the decreased $m_{\Delta O3-\Delta Tmax}$, model simulations estimate that reduced anthropogenic NO$_x$ emissions from 1995 to 2017 have lowered ozone enhancement from low to high temperatures by 6.8 ppbv averaged over the CONUS, significantly reducing the risk of extreme ozone pollution events under high temperatures. Our study illustrates the dependency of ozone-temperature sensitivity on anthropogenic emission levels that should be considered in the future ozone mitigation in a warmer climate.

## 1. Introduction

Surface ozone harms human health and causes loss of crop yields (Feng et al., 2022; Mills et al., 2018; Monks et al., 2015; Turner et al., 2016; Wang et al., 2024). It is chemically generated from its precursors including nitrogen oxides ($NO_x$), volatile organic compounds (VOCs), and carbon monoxide (CO) in the presence of sunlight. The natural sources, chemical kinetics, deposition, and transport of ozone and its precursors are significantly influenced by meteorology and climate (Fiore et al., 2012; Fu and Tian, 2019; Jacob and Winner, 2009; Lu et al., 2019b), shaping the strong sensitivity of surface ozone concentration to meteorological parameters such as temperature. Quantification of ozone-meteorology sensitivity provides a useful tool for predicting daily variation of ozone and for understanding climate-chemistry interactions, yet how anthropogenic emission levels may affect the sensitivity remains unclear. Here, we examine whether long-term anthropogenic control of ozone precursors has changed the response of summertime ozone to daily variations in temperature in the United States (US) and the underlying mechanisms.

High temperature is expected to increase ozone concentrations in polluted environment, through boosting biogenic VOCs (BVOCs) and soil $NO_x$ emissions, accelerating photochemical kinetics of ozone formation, and suppressing ozone dry deposition (Hudman et al., 2012; Lin et al., 2020; Porter and Heald, 2019; Pusede et al., 2015; Romer et al., 2018; Varotsos et al., 2019). In addition, temperature-dependent meteorological parameters, such as solar radiation and humidity, and temperature-related meteorological effects, such as air stagnation, ventilation, and regional transport, can also influence surface ozone level (Kerr et al., 2019; Lu et al., 2019b; Porter and Heald, 2019; Zhang et al., 2022a). Such effects can be reflected in but at the same time complicate the ozone-temperature relationship. Still, temperature is often used as a proxy to synthesize the effects of meteorology and climate on ozone. Previous studies have documented a robust positive ozone-temperature sensitivity in $NO_x$-rich environment, typically defined as the slope of the best-fit line for ozone and temperature ($d[O_3]/dT$), of 2-8 ppbv $K^{-1}$ across the US, Europe, and China (Bloomer et al., 2009; Gu et al., 2020; Ning et al., 2022; Pusede et al., 2014; Sillman and Samson, 1995; Varotsos et al., 2019). The positive $d[O_3]/dT$ also indicates an ozone climate change penalty, *i.e.* future warming may deteriorate ozone air quality in the absence of changes in anthropogenic emission activities (Zhang et al., 2022b). The climate penalty requires additional anthropogenic emission reductions to offset the ozone increase in a warmer climate (Wu et al., 2008; Zanis et al., 2022).

While the overall positive ozone-temperature relationship is well recognized, how ozone-temperature sensitivity has changed remains much less explored. Some studies have reported the weakening of regional ozone-temperature sensitivity in California, the Midwestern US, and the eastern US based on observations, and supposed reduction in local anthropogenic emissions as a possible driver (Bloomer et al., 2009; Hembeck et al., 2022; Jing et al., 2017; Rasmussen et al., 2013). In contrast, Fu et al. (2015) reports large interannual variations in ozone-temperature sensitivity in the southeast US that may be tied to climate variability. Model simulations project a decrease in ozone-temperature sensitivity in future scenarios with lower anthropogenic emissions in the US (Nolte et al., 2021). These studies indicate that the surface ozone-temperature sensitivity has been shifting with significant regional variations in the US, yet an up-to-date view on the long-term and continental-scale

trends is currently missing. In particular, the quantitative assessment of underlying mechanisms driving long-term changes in surface ozone-temperature sensitivity remains rather unclear, limiting the application of this important metric in predicting future ozone evolution.

In this study, we analyze the present-day (2017-2021) and long-term trends (1990-2021) in the summertime surface ozone-temperature relationship in the continental US (CONUS), combining observational monitoring network and state-of-art chemical modeling. We utilize the GEOS-Chem chemical transport model to quantify the role of anthropogenic emission reduction in the long-term trends in the ozone-temperature sensitivity, and investigate the underlying mechanisms. We also examine the benefit of reduced ozone-temperature sensitivity in ozone mitigation during high temperatures that frequently cause severe ozone pollution extremes.

## 2. Materials and Methods

### 2.1 Surface ozone measurement in the US

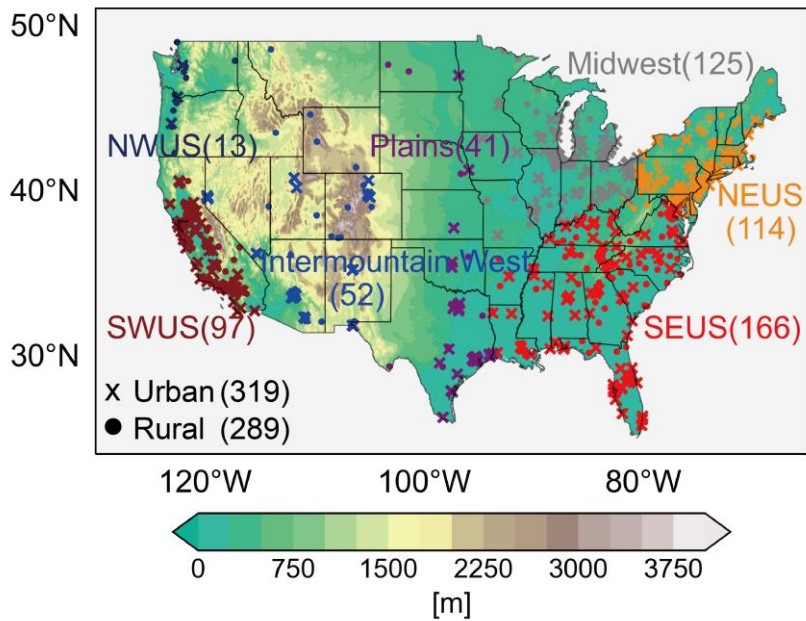

**Figure 1. Locations of the 319 urban sites (crosses) and 289 rural sites (dots) across the continental US used in this study. Sites are categorized in seven regions, including Northwestern US (NWUS), Southwestern US (SWUS), Northeastern US (NEUS), Southeastern US (SEUS), Midwestern US (Midwest), the mountainous western US (Intermountain West), and Central Plains of the US (Plains). The underlying figure shows terrain elevation.**

We obtain hourly measurements of surface ozone concentration from the US Environmental Pollution Agency (EPA) Air Quality System (AQS) data program (https://www.epa.gov/aqs, last access: 10 June 2024). Our study period covers 1990-2021,

in total of 32 years, with a focus on boreal summertime (June, July, August). We derive the daily maximum 8-hour average (MDA8) ozone concentrations from the hourly data, and select sites with valid summertime ozone measurements for at least 24 years (i.e. ≥75%) in the 1990-2021 period and for at least 3 years in 2017-2021 (Text S1). A total of 608 sites are selected, including 319 urban sites and 289 rural sites (based on EPA categorization). We follow previous studies to categorize the sites into seven geographic areas (Nolte et al., 2021; Rasmussen et al., 2012), including the Northwestern US (NWUS), Southwestern US (SWUS), Northeastern US (NEUS), Southeastern US (SEUS), Midwestern US (Midwest), the mountainous western US (Intermountain West), and Central Plains of the US (Plains) (Figure 1).

## 2.2 Temperature data

The AQS dataset also provides surface temperature measurements that could be ideally used in quantifying the ozone-temperature relationship at individual sites. However, the temperature measurement is largely missing, with only 170 sites (<30% of the total 608 sites selected for analysis) providing long-term (at least 24 years) records, which is insufficient to support our analysis. Here we use the gridded (0.5°×0.625°) data of temperature at 2 meters above the ground from the Modern-Era Retrospective analysis for Research and Applications (Version 2, MERRA-2) dataset (Gelaro et al., 2017), which consistently serves as input for the GEOS-Chem chemical transport model (Section 2.4). We align the gridded temperature data with in-situ ozone measurement based on the coordinates of individual sites. Evaluation of the MERRA-2 gridded data with in-situ measurements of temperature at available sites shows an excellent agreement between the two with a mean bias (MB) of 0.3-1.0 K and the correlation coefficient ($r$) of 0.96-0.98 for years after 2000, however, the two datasets have slightly larger disparities in the earliest part of our study period (e.g. MB=0.5 K, r=0.87 for year 1990) (Figure S1). We also compare temperature trends from MERRA-2 with observations over the period 1990-2021 (Table S1). While the overall trends are consistent, there are notable overestimation (e.g. NEUS, Plains) and underestimation (e.g. SEUS and SWUS) in different regions, which may lead to biases in interpreting the observed ozone-temperature sensitivity (as observed ozone variation responds to "true" air temperature).

## 2.3 Definition of ozone-temperature sensitivity

Our goal is to examine the response of summertime MDA8 ozone concentration to the variation in daily maximum temperature ($T_{max}$) across the US, and the trends in such response from 1990 to 2021. We use $T_{max}$ instead of daytime temperature or mean temperature as strong correlation coefficients between MDA8 ozone and $T_{max}$ have been revealed in previous studies (e.g. Steiner et al., 2010; Fu et al., 2015). Ozone levels in the US have experienced significant decreasing trends since 1980s due to anthropogenic emission control measures (Gaudel et al., 2018; Kim et al., 2006; Lin et al., 2017; Simon et al., 2015). The higher ozone concentration in earlier years may obfuscate the long-term trends in ozone-temperature sensitivity, if the ozone-temperature sensitivity were derived by the raw measurements. Therefore, we first subtract the monthly-mean MDA8 ozone concentration and $T_{max}$ from each daily record to derive their daily anomaly ($\Delta O_3$ and $\Delta T_{max}$) at

individual sites for each year. This process allows us to remove the seasonal (monthly) influences and also the 1990-2021 trends in ozone concentration and temperature. We then define the summertime ozone-temperature sensitivity ($m_{\Delta O3\text{-}\Delta Tmax}$) at individual sites as the slope of the best-fit line of daily $\Delta O_3$ versus $\Delta T_{max}$. Fu et al. (2015) also applied similar process to quantify ozone-temperature sensitivity across the southeast US. We calculate the mean values of $m_{\Delta O3\text{-}\Delta Tmax}$ over the sites across the contiguous US (CONUS) or individual regions to represent the regional-mean ozone-temperature sensitivity. Trends in $m_{\Delta O3\text{-}\Delta Tmax}$ over each site are estimated using the linear regression method, with a 5-year smoothing average applied to the yearly $m_{\Delta O3\text{-}\Delta Tmax}$ to filter the interannual variability. The trends of the mean $m_{\Delta O3\text{-}\Delta Tmax}$ values across the sites are used to represent regional mean trends in the ozone-temperature sensitivity.

## 2.4 GEOS-Chem model simulation

We use the GEOC-Chem version 11-02-rc chemical transport model (available at http://geos-chem.org, last access: 10 June 2024; Bey et al., 2001) to interpret summertime ozone-temperature sensitivity and its trend in the United States. The GEOS-Chem model is driven by MERRA-2 assimilated meteorological data. We conduct simulations over the North America nested-grid domain (140°-40° W, 10°-70° N) at the horizontal resolution of 0.5°(latitude) × 0.625°(longitude). The global simulations at 2° × 2.5° resolution providing the boundary conditions were configured consistently with the nested simulations (simulation time, chemical schemes, emission inventory, etc., see discussions below). GEOS-Chem model describes a state-of-art ozone–$NO_x$–VOCs–aerosol–halogen tropospheric chemistry scheme, and also includes online calculation of emissions, dry and wet depositions of gases and aerosols. Anthropogenic emissions in this study are from the Community Emissions Data System (CEDS v-2021-04-21), in which the interannual variability in the US emissions are scaled based on the US National Emissions Inventory (US NEI) (McDuffie et al., 2020). The CEDS inventory indicates a significant decrease in anthropogenic $NO_x$, NMVOCs, CO emissions over the CONUS of 62.5, 70.8, 48.0% respectively from 1995-2017.

GEOS-Chem is capable of simulating the temperature's influences on ozone through chemical kinetics, natural emissions, transport, and dry deposition. Chemical kinetics in GEOS-Chem are modularized based on the Jet Propulsion Laboratory (JPL) and International Union of Pure and Applied Chemistry (IUPAC) scheme (IUPAC, 2011; Sander et al, 2013), with temperature input from the hourly MERRA-2 reanalysis data. GEOS-Chem also includes online calculation of temperature-dependent natural emissions. Biogenic emissions are parameterized following The Model of Emissions of Gases and Aerosols from Nature (MEGAN version v2.1) algorithm (Guenther et al., 2012), in which biogenic emissions are calculated based on temperature, solar radiation, leaf area index (LAI), and other parameters. Biogenic emissions increase exponentially with temperature, but emissions of some BVOCs are inhibited at higher temperatures. Soil $NO_x$ emissions are calculated based on nitrogen availability in soil, edaphic conditions such as soil temperature and moisture, and other gridded parameters such as vegetation type using the Berkeley-Dalhousie Soil $NO_x$ Parameterization (BDSNP) as described in Hudman et al. (2012). Lightning $NO_x$ emissions are parameterized based on cloud-top heights with the spatial distribution of flash rates constrained by satellite observations (Murray et al., 2012). Biomass burning emissions are from the BB4CMIP (biomass burning emissions

for CMIP6) inventory (van Marle et al., 2017), in which emissions after 1997 are consistent with the Global Fire Emissions Database version 4 (GFED4) inventory (van der Werf et al., 2017). However, temperature's impacts on anthropogenic $NO_x$ and VOCs emissions (Liu et al., 2024; Wu et al., 2024) are not considered in our simulation.

Dry deposition of both gas and aerosols is calculated online based on the resistance-in-series algorithm (Wesely, 1989). Surface temperature influences deposition velocity through a stomatal resistance term, which remains low within normal temperatures (e.g. 10-30 °C) but rises at two extremes (below 0 °C and above 40 °C) (Porter and Heald, 2019), contributing to local ozone increases at high temperatures. Wet deposition for water-soluble aerosols and gases in GEOS-Chem is described by Liu et al. (2001) and Amos et al. (2012). $NO_x$ and ozone have low solubility, but wet deposition of $NO_x$ oxidation products may further influence ozone. We do not separately consider temperature's indirect influences on ozone through wet deposition processes in this study.

Model simulations are summarized in Table 1. We conduct a BASE simulation for July for every two years from 1995 to 2017, with one-month simulation (June) as model spin-up for both the global and regional simulations. We do not extend the simulation to earlier or later years due to lack of reliable anthropogenic emission inventory by the time when this study was designed. The initial chemical fields are close to conditions for July 2005 (the same initial fields used for each set of sensitivity experiments). The one-month spin-up time can be considered sufficient in this case as the ozone in the urban boundary layer typically has a lifetime ranging from hours to days. However, it may be short for ozone in the free troposphere where ozone has a lifetime of orders of weeks (Monks et al., 2015). To demonstrate this, we conducted an additional set of experiments, starting with a global simulation at 2°×2.5° resolution from 1st January 2017 to 1st August 2017. The global simulation on 1st June 2017 was then interpolated into the high-resolution nested grid to drive the high-resolution simulation from 1st June 2017 to 1st August 2017. A comparison of surface MDA8 ozone concentrations and ozone-temperature sensitivity between the two sets of simulations is shown in Figure S2. We find that the differences between the simulations with 1-month and 6-month spin-up times had only minor impacts on ozone concentrations and $m_{\Delta O3-\Delta Tmax}$. The average differences between the two simulations were only 0.3% for ozone concentrations and 2.3% for $m_{\Delta O3-\Delta Tmax}$, with high spatial consistency (r > 0.99). This confirms that using a 1-month spin-up time for the simulation should not affect the analysis and conclusions. However, for specific regions, more noticeable differences in ozone concentrations and $m_{\Delta O3-\Delta Tmax}$ exist between the two simulations. A longer spin-up time is favorable for generating global chemical fields when sufficient computational resources are available. The BASE simulation applies the yearly-varied anthropogenic emissions and includes all the abovementioned temperature-dependent mechanisms. We then conduct two simulations for the same 1995-2017 period, but in which the domestic anthropogenic $NO_x$ (1995E) or VOCs (1995EAVOCs) emissions in US are fixed to 1995 level.

We conducted 14 additional sets of sensitivity experiments to explore the role of different mechanisms in the ozone-temperature sensitivity and its trend. First, we separate the effect of temperature on ozone through direct and indirect effects. Here, the temperature-direct effect is defined as the effect directly parameterized with temperature in GEOS-Chem, including natural emissions of BVOCs and soil $NO_x$, the chemical kinetics, dry deposition, and other mechanisms that may have minimal impacts on ozone. In comparison, temperature-indirect effect is defined as the effect not directly parameterized with

temperature but is also influenced or reflected by temperature, for example humidity, radiation, and transport. The simulation strategy is to remove the daily variation of temperature (while keeping the diurnal cycle) and its influence on ozone daily variations. For this purpose, we generate the mean diurnal cycle of temperature averages over all 31 days in July 2017 at each grid cell. We then feed this normalized temperature data into the calculations of the GEOS-Chem (FTEMP). As such, the FTEMP simulation identifies the indirect effect of temperature on ozone. Comparison of the FTEMP simulation and BASE simulation yields a quantitative assessment of the direct effect.

For the temperature-direct effect, we further follow Porter and Heald (2019) to explore the role of four temperature-dependent mechanisms on the ozone-temperature sensitivity. These four mechanisms are BVOCs emissions, soil $NO_x$ emissions, thermal decomposition of peroxyacetyl nitrate (PAN, whose decomposition is strongly correlated to temperature), and dry deposition. We feed the normalized temperature data (remove daily variation but keep diurnal cycle) into the calculations of all or each of the four temperature-dependent mechanisms in the GEOS-Chem. For the temperature-indirect effect, we additionally examine the role of transport in the ozone-temperature sensitivity. This is done by generating a meteorological field that retains only the daily variation of three-dimensional wind field and boundary layer height (PBLH) and removes the daily variation of all other meteorological elements, which is used into the GEOS-Chem (TRANS). Interpretation of these sensitivity simulations are summarized in Table S2.

We conduct the above simulations at both the 2017 and 1995 emission level, allowing us to explore the role of these mechanisms in the changes in ozone-temperature sensitivity with anthropogenic $NO_x$ emissions reduction, which has not been addressed in previous modeling studies. Except for the BASE, 1995E, and 1995EAVOCs simulation, other simulations are only conducted for year 2017 (the latest year with available anthropogenic emission inventory when the simulations were conducted) as sensitivity tests.

**Table 1 Configurations of Model Simulations**

| Cases | Simulation time | Description |
|---|---|---|
| **BASE** | July,1995-2017 (biennially) | Default simulation with yearly-varied anthropogenic emissions and all temperature-dependent mechanisms |
| **1995E** | Same as BASE | Same as BASE, but anthropogenic $NO_x$ emissions fixed in 1995 |
| **1995EAVOCs** | Same as BASE | Same as BASE, but anthropogenic VOCs emissions fixed in 1995 |
| **BASE-FTEMP** | July, 2017 | Same as BASE, but with normalized temperature field in the model |
| **1995E-FTEMP** | July, 2017 | Same as 1995E, but with normalized temperature field in the model |
| **BASE-TRANS** | July, 2017 | Same as BASE, but with normalized all meteorological elements |

| | | except three-dimensional wind field and PBLH |
|---|---|---|
| **1995E-TRANS** | July, 2017 | Same as 1995E, but with normalized all meteorological elements except three-dimensional wind field and PBLH |
| **BASE-F4PATHS** | July, 2017 | Same as BASE, but remove four mechanisms temperature dependence by normalized temperature for BVOCs emissions, Soil $NO_x$ emissions, PAN decomposition, and dry deposition |
| **1995E-F4PATHS** | July, 2017 | Same as 1995E, but remove four mechanisms temperature dependence by normalized temperature for BVOCs emissions, Soil $NO_x$ emissions, PAN decomposition, and dry deposition |
| **BASE-FBVOC** | July, 2017 | Same as BASE, but remove BVOCs temperature dependence by normalized temperature for biogenic VOC emissions |
| **1995E -FBVOC** | July, 2017 | Same as 1995E, but remove BVOCs temperature dependence by normalized temperature for biogenic VOC emissions |
| **BASE-FSNO$_x$** | July, 2017 | Same as BASE, but remove Soil $NO_x$ temperature dependence by normalized temperature for soil $NO_x$ emissions |
| **1995E-FSNO$_x$** | July, 2017 | Same as 1995E, but remove Soil $NO_x$ temperature dependence by normalized temperature for soil $NO_x$ emissions |
| **BASE-FPAN** | July, 2017 | Same as BASE, but remove PAN temperature dependence by normalized temperature for PAN decomposition |
| **1995E-FPAN** | July, 2017 | Same as 1995E, but remove PAN temperature dependence by normalized temperature for PAN decomposition |
| **BASE-FDEP** | July, 2017 | Same as BASE, but remove dry deposition temperature dependence by normalized temperature for dry deposition |
| **1995E-FDEP** | July, 2017 | Same as 1995E, but remove dry deposition temperature dependence by normalized temperature for dry deposition |

## 3.   Results

### 3.1  Present-day level and trends of summertime ozone-temperature sensitivity in the continental US

205

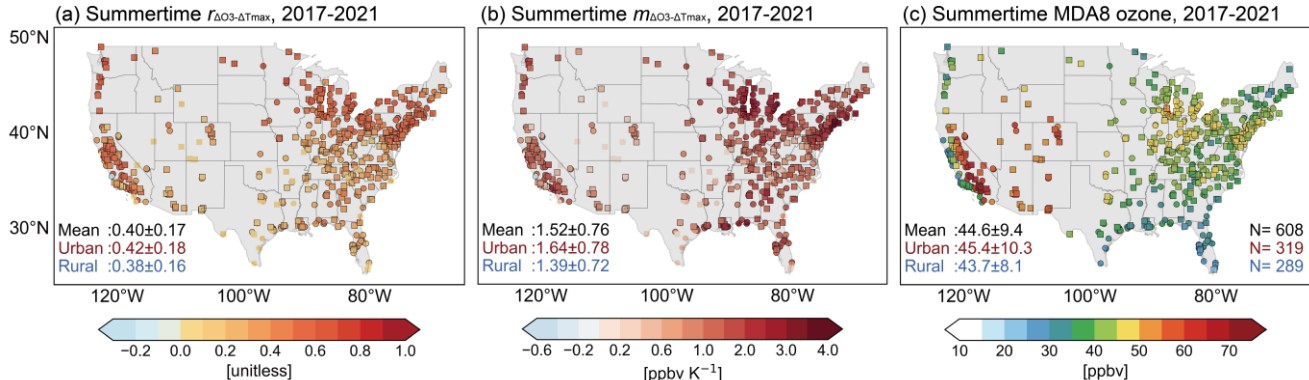

**Figure 2. Present-day summertime ozone concentrations and ozone-temperature sensitivity in the continental US. (a,b) Distribution of summertime (June, July, August) $r_{\Delta O3-\Delta Tmax}$ and $m_{\Delta O3-\Delta Tmax}$ at individual sites averaged in 2017-2021. Black borders indicate sites with a p-value<0.01 for $r_{\Delta O3-\Delta Tmax}$ or $m_{\Delta O3-\Delta Tmax}$. (c) Summer mean MDA8 ozone concentrations at individual sites. Urban sites are represented by circle and rural sites by square. Mean values and standard deviations over the sites are shown in the inset.**

Figure 2(a) presents the widespread positive correlation coefficients between summertime daily MDA8 ozone and $T_{max}$ ($r_{\Delta O3-\Delta tmax}$) across the CONUS sites. 604 out of the total 608 sites are showing positive $r_{\Delta O3-\Delta Tmax}$ (568 sites with p-value<0.01) in the present-day (2017-2021), with a mean $r_{\Delta O3-\Delta Tmax}$ value of 0.40±0.17 (mean ± standard deviation across the sites) averaged over all sites. Urban sites show slightly higher $r_{\Delta O3-\Delta Tmax}$ values than rural sites. Figure 2(b) shows that the present-day mean $m_{\Delta O3-\Delta Tmax}$ (see Section 2.3 for the definition) values averaged for the 608 sites are 1.52±0.76 ppbv $K^{-1}$, with the $m_{\Delta O3-\Delta Tmax}$ values at urban sites higher by 18% than those averaged for the rural sites (1.64±0.78 versus 1.39±0.72 ppbv $K^{-1}$). These results reflect the expected ozone increases with temperature in $NO_x$-rich environment, which are more commonly found in urban than rural areas.

We find distinct variability in the spatial distributions of both $r_{\Delta O3-\Delta Tmax}$ and $m_{\Delta O3-\Delta Tmax}$ (Figure 2, Table S3). The Midwest and NEUS regions show the highest mean $m_{\Delta O3-\Delta Tmax}$ values reaching 2.05±0.62 ($r_{\Delta O3-\Delta Tmax}$=0.50±0.12) and 1.99±0.65 ppbv $K^{-1}$ ($r_{\Delta O3-\Delta Tmax}$=0.52±0.09), followed by NWUS with mean $m_{\Delta O3-\Delta Tmax}$ of 1.54±0.38 ppbv $K^{-1}$ ($r_{\Delta O3-\Delta Tmax}$=0.63±0.07). The Intermountain West and Plains region show the lowest mean $m_{\Delta O3-\Delta Tmax}$ of less than 1.1 ppbv $K^{-1}$ in both urban and rural sites with mean $r_{\Delta O3-\Delta Tmax}$ lower than 0.26, indicating daily ozone variation in this region is not strongly affected by temperature. We also find that the spatial distribution of ozone-temperature sensitivity does not follow that of the MDA8 ozone level (Figure 2c), as the highest summertime MDA8 ozone concentrations are over the SWUS and Intermountain West Regions. The higher $m_{\Delta O3-\Delta Tmax}$ in the NEUS and Midwest regions than in other regions may reflect the stronger daily variation of ozone due to rapid shift of synoptic patterns (e.g. mid-latitude cyclones) in this region during summer (Leibensperger et al., 2008). Additionally, changes in other mid-latitude dynamic systems, such as meridional movement by the mid-latitude jet, also play a significant role in shaping the regional ozone-temperature sensitivity (Barnes and Fiore, 2013; Kerr et al., 2020; Zhang et al.,

2022c). We observe a decreasing gradient in both $r_{\Delta O3\text{-}\Delta Tmax}$ and $m_{\Delta O3\text{-}\Delta Tmax}$ from north to south in the eastern United States, which aligns with previous findings (Camalier et al., 2007; Tawfik and Steiner, 2013). This observed north-to-south shift may be related to the transition in land-atmosphere coupling mechanisms due to soil moisture limitations in the southern regions (Tawfik and Steiner, 2013). The low $m_{\Delta O3\text{-}\Delta Tmax}$ in the Intermountain region largely reflects the strong background ozone influences (including stratospheric intrusion, long-range transport of wildfire or anthropogenic plumes) instead of local photochemical production (Jaffe et al., 2018; Zhang et al., 2014). These background sources may contribute to high ozone there but are not directly modulated by local temperature.

Previous studies report a decrease in ozone concentration at extreme high temperature over the US (Shen et al., 2016; Steiner et al., 2010). Here we investigate how the suppression of ozone concentration influences the overall ozone-temperature sensitivity. We identify occurrences of ozone suppression and the critical temperature (i.e. beyond which ozone increases are suppressed) at individual sites every year following the criteria described in Ning et al (2022). We find that while ozone suppression at extreme high temperature can be detected at 477 out of 608 sites in 2017-2021, excluding data above the critical temperature only changes the present-day mean $m_{\Delta O3\text{-}\Delta Tmax}$ by 2.6%. It indicates that such phenomenon does not change the overall positive ozone-temperature sensitivity.

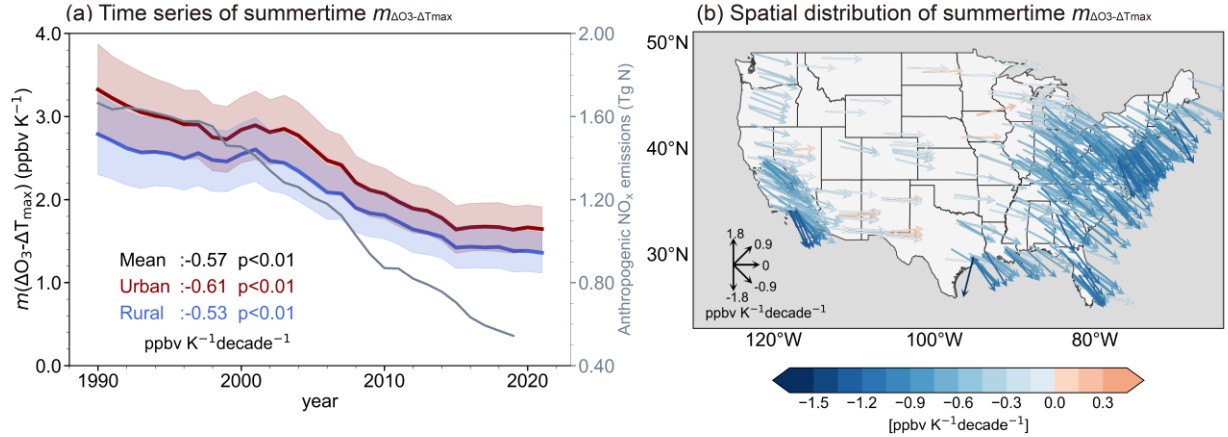

**Figure 3. Observed decrease in summertime ozone-temperature sensitivity in 1990-2021. (a)** Time series of the summertime $m_{\Delta O3\text{-}\Delta Tmax}$ averaged over the CONUS sites, with a 5-year smoothing average applied to the yearly $m_{\Delta O3\text{-}\Delta Tmax}$ to filter the interannual variability. $m_{\Delta O3\text{-}\Delta Tmax}$ for urban and rural sites are shown in red and blue lines, respectively. Shaded areas represent the range of mean values ± 30% of standard deviation across the sites. The CONUS $m_{\Delta O3\text{-}\Delta Tmax}$ trends are shown inset. Anthropogenic NO$_x$ emissions in the CONUS are shown in grey line. **(b)** Spatial distributions of long-term trends in $m_{\Delta O3\text{-}\Delta Tmax}$ in 1990-2021 across the US. Only sites with $r_{\Delta O3\text{-}\Delta Tmax}$ p-values<0.01 with for $r_{\Delta O3\text{-}\Delta Tmax}$ are shown. Both directions and colors of the vectors indicate the $m_{\Delta O3\text{-}\Delta Tmax}$ trends.

The present-day (2017-2021) ozone-temperature sensitivity is lower than the reported values for earlier years (i.e. 2-7 ppbv K$^{-1}$ reported in 2000 (Bloomer et al., 2009; Fu et al., 2015; Rasmussen et al., 2013), though different definitions of ozone-sensitivity sensitivity were applied), suggesting that the ozone-temperature sensitivity may have experienced significant reduction in recent decades. Figure 3 illustrates this feature from long-term observations in 1990-2021. We find in Figure 3a that mean $m_{\Delta O3-\Delta Tmax}$ for the CONUS decreased by 50% from 3.0 ppbv K$^{-1}$ in 1990 to 1.5 ppbv K$^{-1}$ in 2021 with a mean decreasing rate of -0.57 ppbv K$^{-1}$ per decade (p<0.01). $m_{\Delta O3-\Delta Tmax}$ over the CONUS urban sites was higher than rural sites by 0.50 ppbv K$^{-1}$ in the early 1990s. However, urban sites exhibit a faster decline rate of $m_{\Delta O3-\Delta Tmax}$ (-0.61 ppbv K$^{-1}$ per decade, p<0.01) compared to rural (-0.53 ppbv K$^{-1}$ per decade, p<0.01), narrowing the disparity in $m_{\Delta O3-\Delta Tmax}$ between the two. At the same time, the mean $r_{\Delta O3-\Delta Tmax}$ decreased from 0.51 in 1990 to 0.40 in 2021 (Figure S3). The significant decrease in both $m_{\Delta O3-\Delta Tmax}$ and $r_{\Delta O3-\Delta Tmax}$ all imply a much weaker response of ozone to temperature in present-day compared to that in three decades ago. While some studies have shown observed decreases in some regions (e.g. California as described in Steiner et al. (2010)), such significant decreases of ozone-temperature sensitivity over the CONUS have not been presented in previous studies to the best of our knowledge.

The decreasing trends in $m_{\Delta O3-\Delta Tmax}$ are widespread across the CONUS sites (Figure 3b), but spatial and temporal variabilities exist. 419 sites (69%) out of the total 608 sites are showing negative trends with p<0.01(492 sites with p<0.05). The largest decreases are in the NEUS region, where $m_{\Delta O3-\Delta Tmax}$ values exceeded 4.3 ppbv K$^{-1}$ in the 1990s but have steadily decreased by -0.81 ppbv K$^{-1}$ per decade, reaching 1.8 ppbv K$^{-1}$ in 2021. The SWUS region also shows a large decrease in $m_{\Delta O3-\Delta Tmax}$ by -0.60 ppbv K$^{-1}$ per decade(p<0.01). A distinct feature in the SWUS is the notably high urban-rural disparity in $m_{\Delta O3-\Delta Tmax}$ (4.7 versus 1.9 ppbv K$^{-1}$) in the early 1990s (Figure S4), but this disparity has been significantly reduced as urban sites exhibit a much larger $m_{\Delta O3-\Delta Tmax}$ trend (-0.88 ppbv K$^{-1}$ per decade, p<0.01) than rural sites (-0.34 ppbv K$^{-1}$ per decade, p<0.01), particularly in early 1990s. The SEUS and Midwest regions also show decreases in $m_{\Delta O3-\Delta Tmax}$ with a mean rate of -0.62 and -0.52 ppbv K$^{-1}$ per decade. However, we notice an increase of $m_{\Delta O3-\Delta Tmax}$ in 1990-2000 for the SEUS region and in 1999-2005 for the Plains region (Figure S4). The increase in ozone-temperature sensitivity in these two regions explains the $m_{\Delta O3-\Delta Tmax}$ plateau in CONUS during the 1996-2004 period. Fu et al. (2015) attributes the increase ozone-temperature sensitivity in 1990-2000 in the SEUS to variations in regional ozone advection tied to climate variability. This further underscores the significant influence of climate variability on $m_{\Delta O3-\Delta Tmax}$ trends. The region with the smallest $m_{\Delta O3-\Delta Tmax}$ trends is the Intermountain region (-0.08 ppbv K$^{-1}$ per decade).

## 3.2 Simulated long-term trends in ozone-temperature sensitivity and attribution to anthropogenic emission reduction

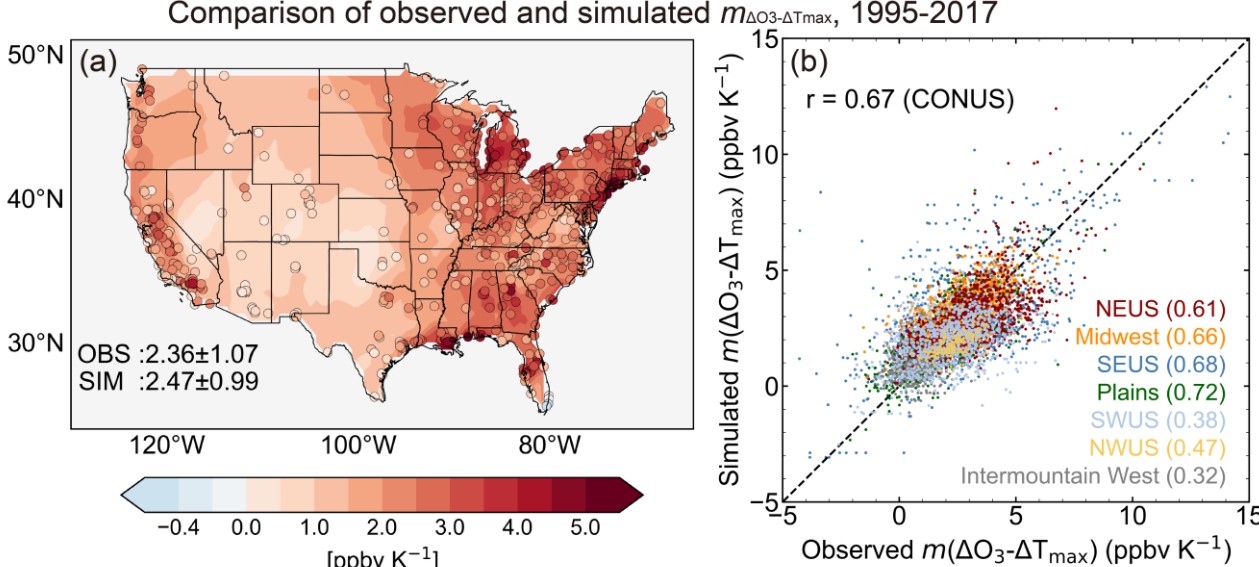

**Figure 4. Evaluation of GEOS-Chem simulated $m_{\Delta O3\text{-}\Delta Tmax}$ in July, 1995-2017. (a) Spatial distributions of the observed (circles) and simulated (from the BASE simulation, shaded) $m_{\Delta O3\text{-}\Delta Tmax}$ during July averaged over 1995 to 2017. (b) Scatterplots of the observed and simulated $m_{\Delta O3\text{-}\Delta Tmax}$ for July in each simulated year from 1995 to 2017. Mean values, standard deviations for the CONUS sites from the observation and GEOS-Chem model, and their correlation coefficients (r) in different regions are shown in the inset.**

We now apply the GEOS-Chem chemical transport model to interpret the trends in ozone-temperature sensitivity over the CONUS. Figure S6 compares the spatial distribution of observed and simulated mean surface MDA8 ozone concentrations in July at the 608 sites averaged for 12 years (1995-2017 biennially). Our GEOS-Chem simulation captures the spatial distributions of surface MDA8 ozone across the CONUS, although showing some high bias of MDA8 ozone of 11 ppbv, as also reported in other surface ozone air quality studies using the GEOS-Chem model (Lu et al., 2019a; Travis and Jacob, 2019). Most importantly, the model largely reproduces the spatial pattern of observed $m_{\Delta O3\text{-}\Delta Tmax}$, with a high correlation coefficient of 0.67 and a small positive mean bias of 0.11 ppbv K⁻¹ (4.7%) at the 608 sites for the monthly $m_{\Delta O3\text{-}\Delta Tmax}$ values at all sites (Figure 4). Table S4 further shows the simulated and observed $m_{\Delta O3\text{-}\Delta Tmax}$ and their correlation coefficients (r) across different periods and regions. The model demonstrates relatively better performance of $m_{\Delta O3\text{-}\Delta Tmax}$ across the CONUS in 2001-2011 compared to other periods, with small mean absolute bias (0.01-0.18 ppbv K⁻¹, 1%-8%) and high correlation coefficients (0.67-0.70). The simulated $m_{\Delta O3\text{-}\Delta Tmax}$ in the Eastern United States (NEUS, SEUS, Midwest, and Plains) is in better agreement with the observed values than in the Western United States, with *r* ranging from 0.50 to 0.76. The above analyses support that the GEOS-Chem model well captures the overall ozone-temperature sensitivity in the period of 1995-2017.

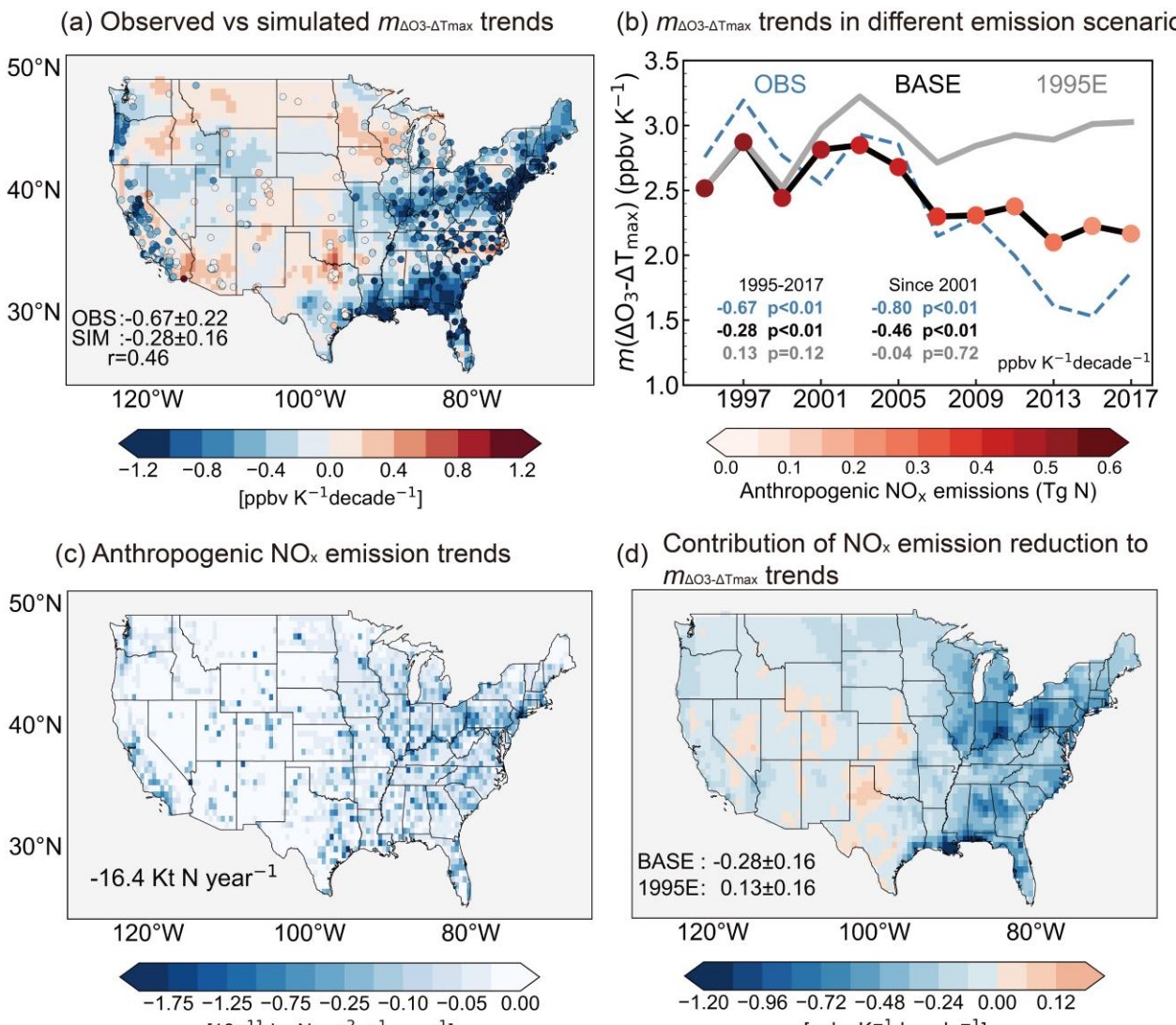

**Figure 5. GEOS-Chem simulated decrease in summertime ozone-temperature sensitivity and the attribution to reduction in anthropogenic NOx emission.** (a) Spatial distributions of the observed (circles) and simulated (from the BASE simulation, shaded) $m_{\Delta O3-\Delta Tmax}$ trends during July from 1995 to 2017. Mean trends±95% confidence level for the CONUS sites from the observation and GEOS-Chem model, and the correlation coefficients (*r*) of $m_{\Delta O3-\Delta Tmax}$ trends between the two are shown inset. (b) Time series of the observed and simulated $m_{\Delta O3-\Delta Tmax}$ in July during 1995-2017 (biennially) at CONUS sites. Results from the BASE simulation and a sensitivity simulation with anthropogenic NOx emissions fixed at 1995 level (1995E) are compared. Colored circles denote the July anthropogenic NOx emissions in the CONUS. (c) Spatial distribution of anthropogenic NOx emission trends during July from 1995 to 2017. Trends are calculated for each model grid. Emissions trends aggregated over the CONUS are insets. (d) Contribution of anthropogenic NOx emissions to $m_{\Delta O3-\Delta Tmax}$ trends, estimated as the difference in the $m_{\Delta O3-\Delta Tmax}$ trend between BASE and 1995E simulations. Mean trends±95% confidence level is shown inset.

Figure 5(a) further compares the observed and GEOS-Chem simulated 1995-2017 trends in $m_{\Delta O3-\Delta Tmax}$ in July across the

CONUS. The following analysis applies biennial data from 1995 to 2017 to align with the GEOS-Chem simulations. The overall observed $m_{\Delta O3\text{-}\Delta Tmax}$ trends in July, as depicted in Figure 5a, are similar to those in June-July-August period as presented in Figure 3b, but there are slight differences at individual sites reflecting the difference in the time frame. We find that driven by yearly-varied meteorological fields and anthropogenic emissions, GEOS-Chem model successfully reproduces the decline of $m_{\Delta O3\text{-}\Delta Tmax}$ across the CONUS, showing a spatial correlation coefficient of 0.46 with observed $m_{\Delta O3\text{-}\Delta Tmax}$ trends (p<0.01). In particular, the model reproduces the much larger $m_{\Delta O3\text{-}\Delta Tmax}$ decreases in the eastern CONUS (the NEUS, Midwest, and SEUS) compared to other regions, consistent with the observations. However, the model has difficulty in capturing the magnitude of observed $m_{\Delta O3\text{-}\Delta Tmax}$ trends. The model shows a mean $m_{\Delta O3\text{-}\Delta Tmax}$ trend of -0.28 ppbv K$^{-1}$ per decade over the CONUS that accounts for 42% of the observed trends of -0.67 ppbv K$^{-1}$ per decade. Figure 5b also shows that the model's underestimation of $m_{\Delta O3\text{-}\Delta Tmax}$ trends is primarily attributed to an overestimation of $m_{\Delta O3\text{-}\Delta Tmax}$ from 2013 to 2017 and an underestimation from 1995 to 1999. The consistency between the observed and simulated $m_{\Delta O3\text{-}\Delta Tmax}$ trends also shows regional differences. As shown in Figure S8, the model reproduces the interannual variation of $m_{\Delta O3\text{-}\Delta Tmax}$ well in the Plains and Intermountain West regions, and also captures 65% of the observed trend in the NWUS. However, in other regions, the model only captures less than 50% of the observed $m_{\Delta O3\text{-}\Delta Tmax}$ trends, with either an overestimation in 2013-2017 or underestimation in 1995-1999.

Our GEOS-Chem simulation has successfully reproduced the observed long-term ozone trend averaged over the CONUS (-6.1 ppbv per decade in GEOS-Chem vs -6.5 ppbv per decade in observations) (Table S5). However, capturing the long-term trends in $m_{\Delta O3\text{-}\Delta Tmax}$ can be more challenging than that of ozone concentrations, as it involves the combined uncertainty in temperature data, simulated ozone concentrations, and the parameterization of ozone-temperature response. The underestimation of $m_{\Delta O3\text{-}\Delta Tmax}$ from 1995 to 1999 may be partly attributed to the larger bias in MERRA-2 temperature dataset compared to other periods (Figure S1), and such bias can propagate to the derivation of observed $m_{\Delta O3\text{-}\Delta Tmax}$ based on MERRA-2 dataset. Excluding the 1995, 1997, and 1999 records improve the model's ability in capturing observed $m_{\Delta O3\text{-}\Delta Tmax}$ trends in the CONUS (-0.46 ppbv K$^{-1}$ per decade in GEOS-Chem vs -0.80 ppbv K$^{-1}$ per decade, 58%). In particular, for the NEUS, Midwest, and SWUS, the model's ability to capture observed $m_{\Delta O3\text{-}\Delta Tmax}$ trends improves from 44%, 49%, and 23% to 83%, 66%, and 54%, respectively. The simulated ozone-temperature sensitivity for 2013–2017 shows an overestimation, particularly in the SEUS and Midwest regions (Figure S8). Christiansen et al. (2024) suggested that the CEDS inventory overestimates post-2010 anthropogenic NO$_x$ emissions, especially in the eastern United States, which may lead to overestimation of ozone-temperature sensitivity in these regions. The GEOS-Chem model also misses several pathways in describing the responses of ozone to temperature, such as the responses in anthropogenic emission and land-atmosphere interaction through soil and vegetation. This will be discussed in detail in Section 4. We do not further differentiate the simulated $m_{\Delta O3\text{-}\Delta Tmax}$ trends at urban and rural sites because the model resolution at about 50 km may be too coarse for such separation.

Previous studies have implied reductions of anthropogenic emissions would result in a decrease in the ozone-temperature sensitivity (Bloomer et al., 2009). Here we explicitly test this theory using our sensitivity experiments with anthropogenic NO$_x$ emissions in US fixed at the 1995 level (1995E). Figure 5b shows that once the anthropogenic NO$_x$ emissions were fixed in

1995, the GEOS-Chem simulate no decrease in $m_{\Delta O3-\Delta Tmax}$ (instead a positive trend by 0.13 ppbv K$^{-1}$ per decade averaged over all sites, p=0.12). This implies that the change in anthropogenic NO$_x$ emissions alone decreases $m_{\Delta O3-\Delta Tmax}$ by -0.41 ppbv K$^{-1}$ per decade for all 608 sites, compared to the observed $m_{\Delta O3-\Delta Tmax}$ trend of -0.67 ppbv K$^{-1}$ per decade, and is apparently the dominant driver of the observed decrease in $m_{\Delta O3-\Delta Tmax}$. In comparison, the simulation with only anthropogenic VOCs emissions fixed at the 1995 level shows negligible difference in $m_{\Delta O3-\Delta Tmax}$ compared to the BASE simulation. We note that the difference in $m_{\Delta O3-\Delta Tmax}$ trend between BASE and 1995E simulations is highly consistent with the spatial distribution of anthropogenic NO$_x$ emission trends ($r$=0.40, p<0.01) (Figure 5c), further confirming that NO$_x$ emission reduction is an important driver of the decline in $m_{\Delta O3-\Delta Tmax}$. Figure S8 illustrates that the regions with $m_{\Delta O3-\Delta Tmax}$ being mostly affected by anthropogenic NO$_x$ emission reductions are located in the eastern CONUS (the NEUS, Midwest, and SEUS), while other regions are less affected.

### 3.3 The underlying mechanisms for the decrease in ozone-temperature sensitivity with reduced NO$_x$ emissions

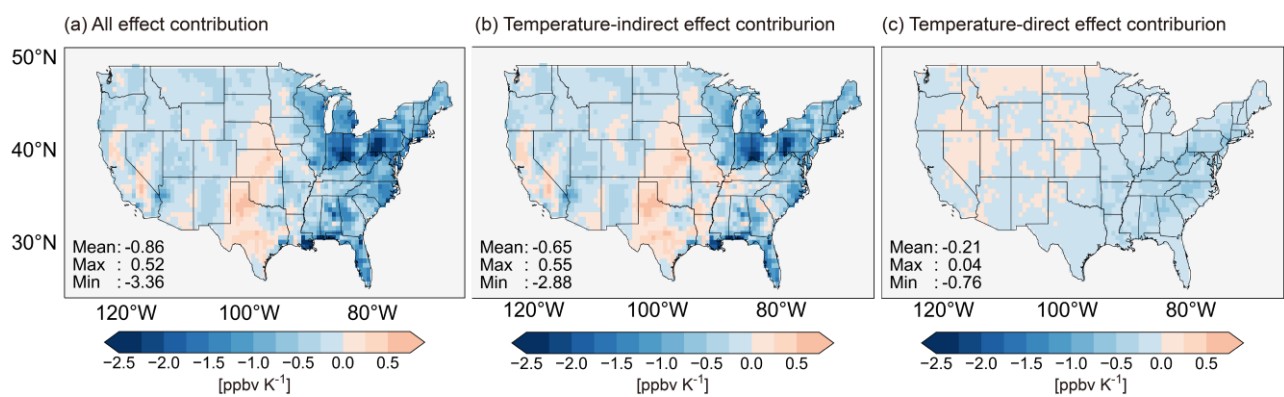

Figure 6. Mechanisms for the decrease in $m_{\Delta O3-\Delta Tmax}$ with anthropogenic NO$_x$ emission reduction. (a) Changes in $m_{\Delta O3-\Delta Tmax}$ due to the difference in anthropogenic NO$_x$ emissions in 2017 and 1995, estimated as the difference in $m_{\Delta O3-\Delta Tmax}$ between the BASE and 1995E simulation for July 2017. (b) The contribution of the temperature-indirect effect to $m_{\Delta O3-\Delta Tmax}$ with changes in anthropogenic NO$_x$ emissions, estimated as the difference of $m_{\Delta O3-\Delta Tmax}$ between BASE-FTEMP and 1995E-FTEMP (Section 2.4). (c) The contribution of the temperature-direct effect, estimated as the difference of $m_{\Delta O3-\Delta Tmax}$ between BASE and 1995E minus the difference between BASE-FTEMP and 1995E-FTEMP. Mean, maximum, and minimum values of the contributions among all CONUS sites are shown inset.

Our analyses above prove that the reduction in anthropogenic NO$_x$ emissions is the dominant driver of the observed long-term decrease in $m_{\Delta O3-\Delta Tmax}$ in the CONUS. We next examine how the changes in anthropogenic NO$_x$ emissions have altered processes controlling ozone's response to temperature. Previous studies have shown that temperature's impacts on surface ozone concentrations involve acceleration of chemical reaction rates, in particular the thermal decomposition of PAN,

increased natural emissions of BVOCs and soil reactive nitrogen, and inhabitation of ozone dry deposition (Lu et al., 2019b; Porter and Heald, 2019; Steiner et al., 2010). Some studies also argued that the temperature-related covariance with other meteorological phenomena such as drought (low humidity), stagnancy, and transport may be more important in determining $m_{\Delta O3\text{-}\Delta Tmax}$ (Kerr et al., 2019; Porter and Heald, 2019; Zhang et al., 2022a, c). Based on these previous studies, we focus on the changes of these impacts on $m_{\Delta O3\text{-}\Delta Tmax}$ with anthropogenic emission reduction in the US.

We illustrate in Figure 6 the simulated changes in $m_{\Delta O3\text{-}\Delta Tmax}$ in July 2017 through temperature-direct effects and temperature-indirect effects associated with the anthropogenic reduction of $NO_x$. Figure 6a shows that the reduction of anthropogenic $NO_x$ emissions from 1995 to 2017 alone decreased $m_{\Delta O3\text{-}\Delta Tmax}$ by 0.86 ppbv K$^{-1}$ in July 2017 (estimated as the difference between the BASE simulation and 1995E simulation). The decreases are larger in the eastern US (including NEUS, Midwest, and SEUS), reaching 1.37, 1.28, and 1.00 ppbv K$^{-1}$, respectively. When the temperature-direct effect is all removed from the GEOS-Chem simulation (Section 2.4), the reduction of anthropogenic $NO_x$ emissions from 1995 to 2017 would decrease $m_{\Delta O3\text{-}\Delta Tmax}$ by 0.65 ppbv K$^{-1}$ in July 2017. It indicates that only a relatively small portion of the decrease in $m_{\Delta O3\text{-}\Delta Tmax}$ (24%, 0.21 ppbv K$^{-1}$ compared with 0.86 ppbv K$^{-1}$) with anthropogenic $NO_x$ reduction can be attributed to the temperature-direct effect (Figure 6c), yet the remaining is explained by temperature-indirect effect. Our results agree with Porter and Heald (2019), which shows that the collinearity between temperature and other meteorological variables played a significant role in determining the overall ozone-temperature relationship. Here, we further demonstrate that the temperature-indirect effect also dominates the decline in ozone-temperature sensitivity with anthropogenic $NO_x$ emission reduction.

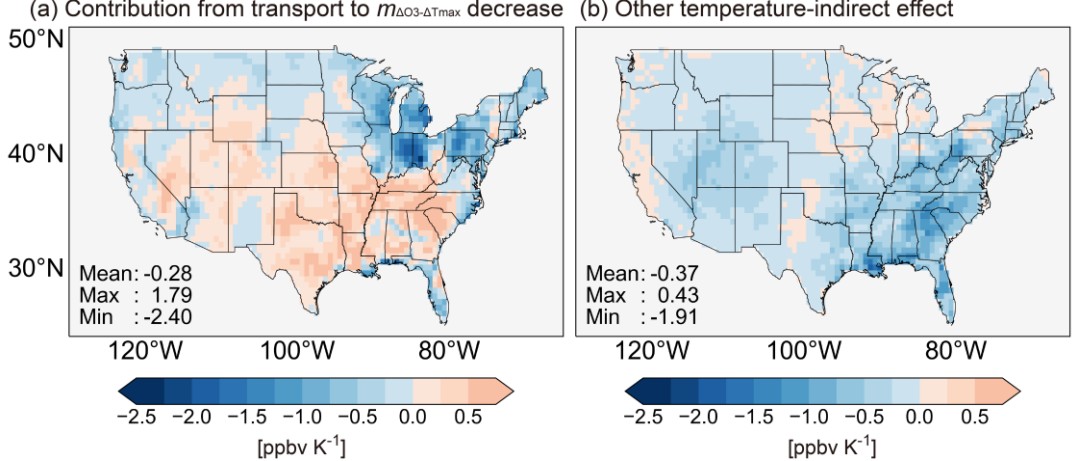

**Figure 7. The different temperature-indirect effects for the decrease in $m_{\Delta O3\text{-}\Delta Tmax}$ with anthropogenic $NO_x$ emission reduction. (a) the contribution of the transport to $m_{\Delta O3\text{-}\Delta Tmax}$ decrease with changes in anthropogenic $NO_x$ emissions, estimated as the difference of $m_{\Delta O3\text{-}\Delta Tmax}$ between BASE-TRANS and 1995E-TRANS (Section 2.4). (b) the contribution of the other temperature-indirect effect, estimated as the difference of $m_{\Delta O3\text{-}\Delta Tmax}$ between BASE-FTEMP and 1995E-FTEMP minus the difference between BASE- TRANS and 1995E- TRANS. Mean, maximum, and minimum values of the contributions among all CONUS sites are shown inset.**

The temperature-indirect effect on ozone mainly includes the influence of temperature-relevant meteorological parameters such as humidity (as an indicator of content of water vapor) and shortwave radiation on ozone photochemistry, and

410 also the effect of transport (including stagnancy and regional transport). We further distinguish the impact of transport (by normalizing all meteorological elements except three-dimensional wind field and PBLH as input in the GEOS-Chem model) and the other indirect effects on the decrease in $m_{\Delta O3-\Delta Tmax}$ with emission reduction. As shown in Figure 7, transport (-0.28 ppbv $K^{-1}$) and other indirect effects (-0.37 ppbv $K^{-1}$) such as humidity and radiation show comparable contribution to the decline in $m_{\Delta O3-\Delta tmax}$, but the spatial patterns show large disparity. The temperature-indirect effect excluding transport (Figure

7b) on $m_{\Delta O3-\Delta tmax}$ shows a more uniform decline with reduced emissions in most regions across the CONUS, with a larger decrease in Southeast US. The radiation received by vegetation in the southeastern United States is highly collinear with temperature and also plays an important role in BVOC emissions (Guenther et al., 2012), which may reflect its potential for ozone formation reduces with the decline in anthropogenic $NO_x$ emissions. In comparison, the transport effect has larger impacts on the $m_{\Delta O3-\Delta Tmax}$ trend (Figure 7a) with reduced $NO_x$ emissions in the northeastern US, where transport has the largest

contribution to the mean $m_{\Delta O3-\Delta Tmax}$ values (Figure S10) as also reported in Kerr et al. (2019). Some studies have demonstrated that changes in mid-latitude weather systems can significantly influence the ozone-temperature sensitivity by affecting pollutant transport (Barnes and Fiore, 2013; Kerr et al., 2020), which could be the underlying mechanism explaining the role of transport in contributing to the decrease of ozone-temperature sensitivity with emission reductions. But we find that these effects cause an increase in $m_{\Delta O3-\Delta Tmax}$ in the southern US in July 2007. Nevertheless, the impact of transport on $m_{\Delta O3-\Delta Tmax}$

largely depends on the transport pattern itself, and it would be more ideally investigated through long-term simulations rather than the one-month study we conducted.

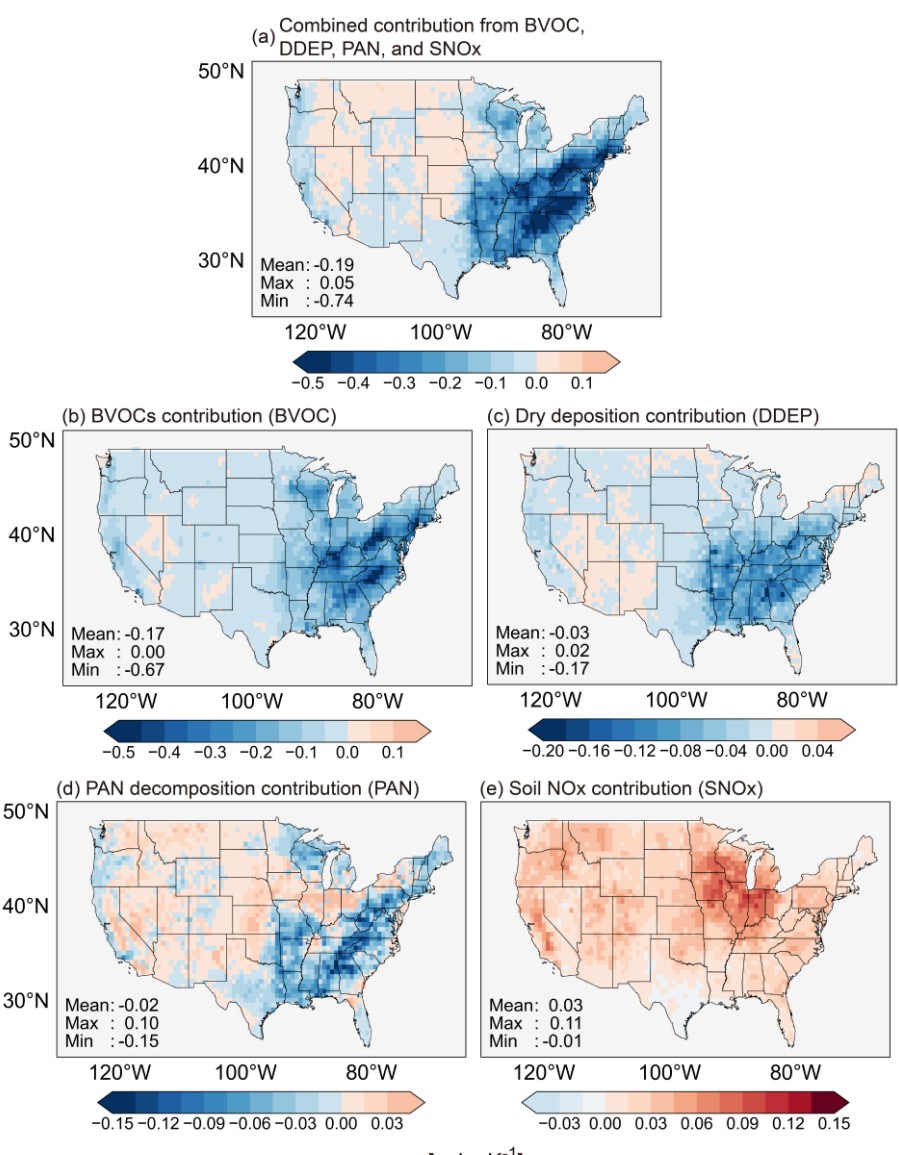

**Figure 8. The different temperature-direct effects for the decrease in $m_{ΔO3-ΔTmax}$ with anthropogenic NO$_x$ emission reduction. (a) Combined contribution of the four temperature-dependent mechanisms (BVOCs emissions, dry deposition, PAN decomposition, and soil NO$_x$ emissions) to $m_{ΔO3-ΔTmax}$ with changes in anthropogenic NO$_x$ emissions, estimated as the difference of $m_{ΔO3-ΔTmax}$ between BASE and 1995E minus the difference between BASE-F4PATHS and 1995E-F4PATHS (Section 2.4). (b)-(e) Individual contribution**
**of BVOCs emissions, dry deposition, PAN decomposition, and soil NO$_x$ emissions) to $m_{ΔO3-ΔTmax}$ with changes in anthropogenic NO$_x$ emissions, respectively. Mean, maximum, and minimum values of the contributions among all CONUS sites are shown inset. Note that the data range of each figure is different.**

The temperature-direct effects on ozone-temperature sensitivity include the explicit impacts of temperature on BVOCs
and soil NOx emissions, chemical kinetics, and dry deposition. Figure 8 shows the additive and individual impacts from the

four temperature-dependent mechanisms (BVOCs, dry deposition, PAN decomposition, and soil $NO_x$) on the decrease in $m_{\Delta O3\text{-}\Delta tmax}$ decreases with reduced $NO_x$ emissions. Comparison of Figure 8a with Figure 6c shows that the contribution of the four temperature-dependent mechanisms contributes to almost all of the $m_{\Delta O3\text{-}\Delta tmax}$ decreases attributable to temperature-direct effect (-0.19 ppbv $K^{-1}$ versus -0.21 ppbv $K^{-1}$).

We find that the ozone-temperature sensitivity contributed by BVOCs emissions has significantly reduced with anthropogenic emission control (Figure 8b). In July 2017, BVOCs emissions alone would have contributed to ozone-temperature sensitivity by 0.2 ppbv $K^{-1}$ if anthropogenic emissions had remained at 1995 levels, with a particularly large contribution of 0.5 ppbv $K^{-1}$ over the parts of eastern US where anthropogenic $NO_x$ emissions are high and ozone formation is sensitivity to VOCs emissions (Figure S11d). However, with anthropogenic $NO_x$ emission decreased to the 2017 level, the

contribution of BVOCs emissions decreases to 0.03 ppbv $K^{-1}$ averaged over the CONUS sites and -0.01 ppbv $K^{-1}$ averaged over the SEUS region (Figure S11c). This suggests that the reduction in anthropogenic $NO_x$ emission has shifted the ozone formation regime to a less VOCs-sensitive state, in which ozone concentrations are much less sensitive to increased BVOCs at high temperatures. Ozone-temperature sensitivity contributed by dry deposition also reduced by -0.03 ppbv $K^{-1}$ averaged over the CONUS sites with anthropogenic emission reduction (Figure 8c).

The thermal decomposition of PAN contributes to 0.43 ppbv $K^{-1}$ of the overall $m_{\Delta O3\text{-}\Delta Tmax}$ over the CONUS (Figure S11g), with larger contribution of 0.7 ppbv $K^{-1}$ over the eastern US states. This is also consistent with Porter and Heald (2019), which shows the PAN decomposition explains a large fraction of the ozone-temperature sensitivity compared to other mechanisms such as BVOCs emissions and dry deposition. The PAN concentrations averaged over the CONUS decrease by 27% with the reduction in anthropogenic $NO_x$ emissions (Figure S12). Nevertheless, $m_{\Delta O3\text{-}\Delta Tmax}$ contributed by PAN

decomposition only shows minor change with the reduction in anthropogenic $NO_x$ emission of -0.02 ppbv $K^{-1}$ averaged over the CONUS (Figure 8d), reflecting the offset between $m_{\Delta O3\text{-}\Delta Tmax}$ increase in the central and western US and decrease in the eastern US. A possible reason is that, with the reduction of anthropogenic $NO_x$ emissions, ozone formation in the central and western US becomes more $NO_x$-sensitive, as such the decomposition of PAN increases ozone-temperature sensitivity. The decrease in $m_{\Delta O3\text{-}\Delta Tmax}$ contributed by PAN decomposition in the eastern US may mainly reflect the reduction of PAN

concentration with anthropogenic $NO_x$ emission reduction (Figure S12).

Unlike the other mechanisms, $m_{\Delta O3\text{-}\Delta Tmax}$ contributed by the temperature-dependent soil $NO_x$ emissions increases by 0.03 ppbv $K^{-1}$ averaged over the CONUS with anthropogenic $NO_x$ emission reduction. The increase in $m_{\Delta O3\text{-}\Delta Tmax}$ reflects the competitive effect between natural soil (from both natural pool and agricultural fertilizer, but are conventionally categorized as natural sources) and anthropogenic (from fossil fuel) $NO_x$ emissions on ozone formation (Lu et al., 2021; Tan et al., 2023).

Soil emissions become an increasingly important source of nitrogen for ozone formation with decreases anthropogenic $NO_x$ emission levels (Guo et al., 2018; Geddes et al., 2022). As soil emissions are larger at higher temperatures, they contribute to an increasing ozone-temperature sensitivity. The above analysis reveals an increasing importance of soil $NO_x$ emissions in determining ozone-temperature sensitivity in a future with low anthropogenic $NO_x$ emissions.

**3.4 Ozone mitigation benefit through the declined ozone-temperature sensitivity**

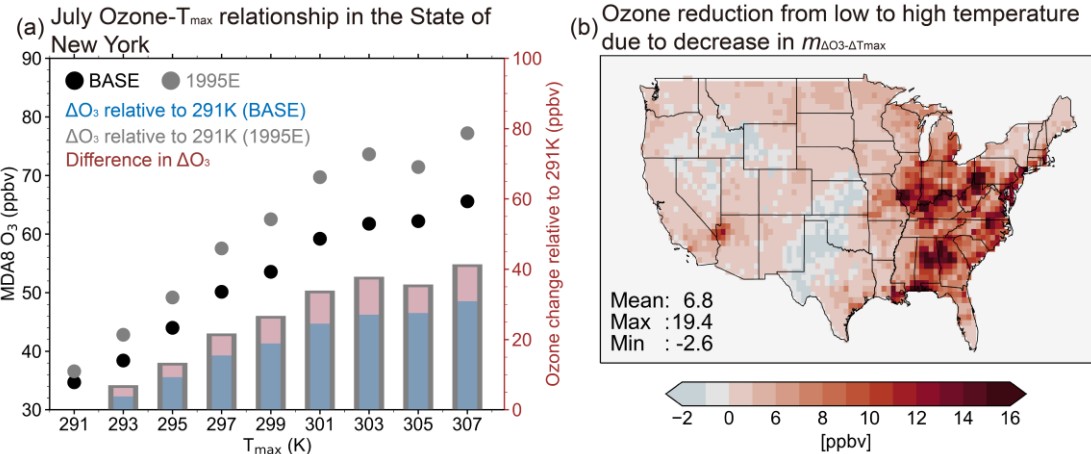

**Figure 9. Decreased $m_{\Delta O3-\Delta Tmax}$ offers ozone mitigation benefit at high temperatures. (a)** Simulated ozone concentration in different $T_{max}$ bins in the State of New York in July 2013, 2015, and 2017. Data are binned to 2K intervals. Results from the BASE simulation (black) and 1995E simulation (grey) are shown. The blue bars represent the ozone enhancement for each temperature bin compared to 291K from the BASE simulation. The bar marked by the grey boarder denotes ozone enhancement for each temperature bin compared to 291K from the 1995E simulation. Thus, the red bar (difference between the grey and blue bars) estimated the decrease in ozone enhancement due to the reduction of anthropogenic emissions from 1995 to 2017. **(b)** Distributions of ozone mitigation benefit in July due to the decreased $m_{\Delta O3-\Delta Tmax}$, estimated as the ozone enhancement from the lowest 0-10% to 90-100% temperatures bins in the 1995E minus that in the BASE at each grid in July (2013,2015 and 2017). Mean, max, and min values for the 608 sites are shown inset.

The significant decrease in $m_{\Delta O3-\Delta Tmax}$ over the CONUS indicates that controlling anthropogenic emission not only reduces the mean ozone levels, but also reduces the response of ozone to temperature. Consequently, this reduction lowers the risk of extreme ozone pollution and associated health damage at high temperatures, presenting an appealing benefit for ozone mitigation. We illustrate the benefit by reducing $m_{\Delta O3-\Delta Tmax}$ in ozone mitigation taking the State of New York as an example, as it has high $m_{\Delta O3-\Delta Tmax}$ and large population exposed to ozone pollution. Figure 9a shows the GEOS-Chem simulated ozone in July for three years (2013, 2015, 2017) at different $T_{max}$ bins. As expected, MDA8 ozone increases with temperature rise. Ozone difference between the highest temperature bin (307K) and the lowest temperature bin (291K) is 31 ppbv in the BASE simulation, comparable to observations (26 ppbv between 307 and 291K). If anthropogenic $NO_x$ emissions were fixed at the 1995 level, however, the predicted ozone difference between the 307K and 291K would be enlarged to 41 ppbv. This means that $NO_x$ emission reductions cause an "additional" ozone concentration reduction of 10 ppbv from 291 to 307 K, as reflected in the significant decline in $m_{\Delta O3-\Delta Tmax}$. Such benefit of reducing $m_{\Delta O3-\Delta Tmax}$ is typically larger at higher temperature. Similar phenomenon can be found in other regions with significant decrease in $m_{\Delta O3-\Delta Tmax}$ (Figure S13).

Figure 9b further quantifies the beneficial effect of anthropogenic emission reduction on ozone mitigation through reducing $m_{\Delta O3-\Delta Tmax}$ over the CONUS. This can be estimated as the suppression of ozone increase between high (90-100[th] percentile of $T_{max}$ in July 2013, 2015, 2017) and low temperature range (the lowest 10[th] percentile of $T_{max}$) due to anthropogenic

NO$_x$ emission reduction from 1995 to 2017. We find that the additional ozone mitigation benefit by reducing $m_{\Delta O3-\Delta Tmax}$ is 6.8 ppbv averaged across the CONUS. The benefit is more pronounced in the eastern US, where emission reductions are more prominent, reaching a maximum of 19.4 ppbv. This benefit significantly reduces the probability of ozone exceedance (MDA8 ozone > 70 ppbv) during high-temperature conditions (above the 90th percentile of T$_{max}$), from 70% (estimated from the 1995E simulation) to 28% (from the BASE simulation). The results show that emission controlled on ozone precursors in the US have effectively reduced the ozone surge at high temperatures across the CONUS, and alleviated the combined health damage in the joint occurrences of heat and ozone extremes, highlighting the importance of continuous anthropogenic emission control on ozone mitigation in a warming future.

## 4. Summary and Discussion

We have estimated in this study the present-day (2017-2021) distributions and long-term (1995-2021) trends in summertime surface ozone-temperature sensitivity in the CONUS, combining observational monitoring network and GEOS-Chem simulations at a resolution of about 50km. We find a clear pattern that the observed $m_{\Delta O3-\Delta Tmax}$ for the CONUS decreased by 50% from 3.0 ppbv K$^{-1}$ in 1990 to 1.5 ppbv K$^{-1}$ in 2021 with a mean decreasing rate of -0.57 ppbv K$^{-1}$ per decade (p<0.01), with urban sites showing faster trends than rural sites (-0.61 vs -0.53 ppbv K$^{-1}$ per decade), indicating a much weaker response of ozone to temperature in present-day compared to that in three decades ago. During the period from 1990 to 2021, anthropogenic NO$_x$ emissions in the United States decreased by approximately 69%, and the eastern United States, where stricter anthropogenic emission controls were implemented, is the core region where ozone-temperature sensitivity has declined the most. The GEOS-Chem simulations driven by year-specific anthropogenic emission inventory and MERRA-2 reanalysis meteorological fields well reproduce the distribution and magnitude of multi-year mean $m_{\Delta O3-\Delta Tmax}$, and capture 42% of the observed trends in $m_{\Delta O3-\Delta Tmax}$ in 1995-2017. The model simulation shows that the decline in anthropogenic NO$_x$ emission over the CONUS is the dominant driver of the $m_{\Delta O3-\Delta Tmax}$ decrease. Mechanically, approximately 76% of the simulated decline in $m_{\Delta O3-\Delta Tmax}$ can be attributed to the temperature-indirect effects arising from the shared collinearity of other meteorological effects (such as humidity, ventilation, and transport) on ozone. The remaining portion explaining the decrease in $m_{\Delta O3-\Delta Tmax}$ with anthropogenic NO$_x$ emission reduction is mostly attributed to four direct temperature-dependent processes, in which $m_{\Delta O3-\Delta Tmax}$ decrease through the pathways of BVOCs emissions, dry deposition, and PAN decomposition (mostly in the eastern US), while soil NO$_x$ emissions increase $m_{\Delta O3-\Delta Tmax}$ with anthropogenic NO$_x$ emission reduction.

Our study illustrates that anthropogenic controls on NO$_x$ emissions have significantly reduced the response of surface ozone concentration to the variation of temperature, offering a compelling advantage for ozone mitigation at high temperatures. The model simulation estimates that the reduction of anthropogenic NO$_x$ emissions from 1995 to 2017 decreases the ozone enhancement from low to high temperatures by 6.8 ppbv on average across the CONUS (reaching 19 ppbv in the part of eastern US). The ozone-temperature sensitivity remains a crucial factor in quantifying the impact of climate on ozone. Our research demonstrates that anthropogenic emission changes not only alleviate current ozone pollution but also help mitigate potential

future increases in ozone concentrations due to climate change. It also indicates the dependency of ozone-temperature sensitivity on anthropogenic emission levels that should be considered in projecting future ozone concentration in a warmer climate.

Nevertheless, there is significant room for improving the ability in capturing the ozone-temperature relationship in the chemical transport model. The GEOS-Chem simulations do not account for the response of anthropogenic $NO_x$ and VOCs emissions to temperature. Recent studies have shown that these emissions can increase simulated regional ozone-temperature sensitivity by up to 7% and 14% (Kerr et al., 2019; Wu et al., 2024). The parameterization of several temperature-dependent processes is limited or even missing in the model. For example, the dry deposition scheme used in this study lacks the temperature response of non-stomatal pathways (Clifton et al., 2020), which could introduce uncertainty in simulated $m_{\Delta O3-\Delta Tmax}$ particularly in vegetation-rich regions such as the southeastern United States. Additionally, according to the BDSNP scheme used in this study, soil $NO_x$ emissions are modeled as an exponential function of temperature between 0 and 30 °C, remaining constant at temperatures above 30 °C. However, some studies have reported continuous increases in soil $NO_x$ emissions at temperatures higher than 30 °C in regions such as California (Oikawa et al., 2015; Wang et al., 2021). The absence of other temperature-dependent natural emissions, such as soil Nitrous acid (HONO) (Tan et al., 2023), may also lead to an underestimation of ozone responses to extreme temperatures in the GEOS-Chem simulations. Uncertainties in the biomass burning emission inventory (Fasullo et al., 2022) limit the accuracy of ozone-temperature sensitivity simulations in fire-impacted regions, such as the mountainous western United States. The 50 km resolution of the model may not fully capture sub-grid meteorological variations, which can play an important role in reproducing extreme conditions at site-level scales. Our study demonstrates that ozone-temperature sensitivity is highly responsive to changes in emissions, emphasizing the importance of more accurate anthropogenic emissions inventory for interpreting the ozone-temperature relationship. Further efforts are needed to enhance the model's ability to capture long-term trends in the ozone response to temperature (including underlying weather conditions and transport patterns), and to better unravel the mechanisms driving the observed ozone-temperature relationship, in particular the role of transport and ventilation.

**Data availability**

The observational data used in this study is open-access as described in the study. The 2 m air temperature from MERRA-2 is available at https://doi.org/10.5067/VJAFPLI1CSIV (Global Modeling and Assimilation Office, 2015). The MERRA-2 reanalysis data is from http://geoschemdata.wustl.edu/ExtData/GEOS_0.5x0.625_NA/MERRA2/ (GMAO, 2024). The anthropogenic emissions data from CEDS is available from https://data.pnnl.gov/dataset/CEDS-4-21-21 (O'Rourke et al., 2021). Data from GEOS-Chem modeling that support the findings of this study is available at https://doi.org/10.5281/zenodo.14250128 (Li et al., 2025). Other supports can be accessed by contacting the corresponding authors (Xiao Lu, luxiao25@mail.sysu.edu.cn).

**Acknowledgement**

This study is supported by the National Key Research and Development Program of China (2023YFC3706104), the Young Elite Scientists Sponsorship Program by CAST (2023QNRC001), and the Fundamental Research Funds for the Central Universities (Sun Yat-sen University, 241gqb004).

**Financial support**

This research has been supported by the National Key Research and Development Program of China (2023YFC3706104), the Young Elite Scientists Sponsorship Program by CAST (2023QNRC001), and the Fundamental Research Funds for the Central Universities (Sun Yat-sen University, 241gqb004).

**Author contribution**

X.L. designed the study. S.L. performed the model simulations and data analyses with significant input from H.L.W.. S.L. and X.L. wrote the manuscript.

**Competing interests.**

The contact author has declared that none of the authors has any competing interests

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
