# Peer review of "Anthropogenic emission controls reduce summertime ozonetemperature sensitivity in the United States"

_EGUsphere, 2024_

## Author Comment (AC1)

**Reviewer #1**

**Comment [1-1]:** This study explores how changing anthropogenic emissions have impacts the O3 sensitivity to temperature from 1990 to 2021 in the United States. The observations demonstrate a diminishing sensitivity, and the authors use GEOS-Chem model simulations to show this is due to decreasing anthropogenic NOx emissions, and that both the direct O3-T mechanisms and indirect (non-T meteorological) factors contribute to this. The study is a very nice example of using a model to interpret an observed result. The study is thorough and well-explained and was a pleasure to read. There are three topics that should be better addressed in the text prior to publication:

**Response [1-1]: We thank the reviewer for the positive and valuable comments. All of them have been implemented in the revised manuscript. Please see our itemized responses below.**

**Comment [1-2]:** A central result of this study is that the model can only reproduce less than half (42%) of the observed trend in the $O_3$-T relationship. The paper attributes the source of the trend in the model, but does not sufficiently discuss the possible reasons and implications of the model missing half of the trend. The authors suggest that biases in the MERRA-2 temperature dataset from 1995-1999 contributes to this, but they show that when excluding this part of the record, the model can still only capture 56% of the trend. On lines 304 & 309 they suggest that this is due to biases in the SWUS region, but neither the region nor the degree of bias seen in Figure 5a seem sufficient to explain the majority of the missing trend. If it is indeed due to the SWUS, the paper should (1) discuss why the model cannot capture this region and (2) show that the model can capture the trend without the SWUS and (3) proceed only with non-SWUS results. If the SWUS can only explain a small part of this model bias, then the paper should discuss what other factors could contribute to the model bias, how this could be further explored in future studies and/or how the model could be improved. The paper brings up the temperature impact on anthropogenic emissions on line 146 – how important might this factor be?

**Response [1-2]: Thank you for pointing this out. We agree that the discussion of the underestimation in simulated $m_{\Delta O3-\Delta Tmax}$ trends should be much strengthened. We find that the model's ability to capture the $m_{\Delta O3-\Delta Tmax}$ trends differs across regions, and the timing of the bias is**

also not consistent. For example, the model underestimates $m_{\Delta O3-\Delta Tmax}$ from 1995 to 1999 mainly in the SWUS and NEUS regions, while overestimations for 2013 to 2017 are seen in the SEUS and Midwest, suggesting that these biases stem from different causes. In the revision, we have analyzed the regional model bias in $m_{\Delta O3-\Delta Tmax}$ in detail, and discuss three main sources of the underestimation of $m_{\Delta O3-\Delta Tmax}$ trends: bias in MERRA-2 temperature data in early periods, overestimations in anthropogenic emissions inventories, and model capability in parameterizing ozone-temperature response. We have added the following discussion to Section 3.2:

"The model shows a mean $m_{\Delta O3-\Delta Tmax}$ trend of -0.28 ppbv/K/decade over the CONUS that accounts for 42% of the observed trends of -0.67 ppbv/K/decade. Figure 5b also shows that the model's underestimation of $m_{\Delta O3-\Delta Tmax}$ trends is primarily attributed to an overestimation of $m_{\Delta O3-\Delta Tmax}$ from 2013 to 2017 and an underestimation from 1995 to 1999. The consistency between the observed and simulated $m_{\Delta O3-\Delta Tmax}$ trends also shows regional differences. As shown in Figure S8, the model reproduces the interannual variation of $m_{\Delta O3-\Delta Tmax}$ well in the Plains and Intermountain West regions, and also captures 65% of the observed trend in the NWUS. However, in other regions, the model only captures less than 50% of the observed $m_{\Delta O3-\Delta Tmax}$ trends, with either an overestimation in 2013-2017 or underestimation in 1995-1999.

Our GEOS-Chem simulation has successfully reproduced the observed long-term ozone trend averaged over the CONUS (-6.1 ppbv/decade in GEOS-Chem vs -6.5 ppbv/decade in observations) (Table S5). However, capturing the long-term trends in $m_{\Delta O3-\Delta Tmax}$ can be more challenging than that of ozone concentrations, as it involves the combined uncertainty in temperature data, simulated ozone concentrations, and the parameterization of ozone-temperature response. The underestimation of $m_{\Delta O3-\Delta Tmax}$ from 1995 to 1999 may be partly attributed to the larger bias in MERRA-2 temperature dataset compared to other periods (Figure S1), and such bias can propagate to the derivation of observed $m_{\Delta O3-\Delta Tmax}$ based on MERRA-2 dataset. Excluding the 1995, 1997, and 1999 records improve the model's ability in capturing observed $m_{\Delta O3-\Delta Tmax}$ trends in the CONUS (-0.46 ppbv/K/decade in GEOS-Chem vs -0.80 ppbv/K/decade, 58%). In particular, for the NEUS, Midwest, and SWUS, the model's ability to capture observed $m_{\Delta O3-\Delta Tmax}$ trends improves from 44%, 49%, and 23% to 83%, 66%, and 54%, respectively. The simulated ozone-temperature sensitivity for 2013–2017 shows an overestimation, particularly in the SEUS and Midwest regions (Figure S8). Christiansen et al. (2024) suggested that the CEDS inventory overestimates post-2010

anthropogenic NO$_X$ emissions, especially in the eastern United States, which may lead to overestimation of ozone-temperature sensitivity in these regions. The GEOS-Chem model also misses several pathways in describing the responses of ozone to temperature, such as the responses in anthropogenic emission and land-atmosphere interaction through soil and vegetation. This will be discussed in detail in Section 4."

Regarding the influence of anthropogenic emissions' response to temperature on ozone-temperature sensitivity, we classified this under model uncertainties and further elaborate on this in section 4: "Nevertheless, there is significant room for improving the ability in capturing the ozone-temperature relationship in the chemical transport model. The GEOS-Chem simulations do not account for the response of anthropogenic NO$_X$ and VOCs emissions to temperature. Recent studies have shown that these emissions can increase simulated regional ozone-temperature sensitivity by up to 7% and 14% (Kerr et al., 2019; Wu et al., 2024). The parameterization of several temperature-dependent processes is limited or even missing in the model. For example, the dry deposition scheme used in this study lacks the temperature response of non-stomatal pathways (Clifton et al., 2020), which could introduce uncertainty in simulated $m_{\Delta O3\text{-}\Delta Tmax}$ particularly in vegetation-rich regions such as the southeastern United States. Additionally, according to the BDSNP scheme used in this study, soil NO$_x$ emissions are modeled as an exponential function of temperature between 0 and 30 °C, remaining constant at temperatures above 30 °C. However, some studies have reported continuous increases in soil NO$_x$ emissions at temperatures higher than 30 °C in regions such as California (Oikawa et al., 2015; Wang et al., 2021). The absence of other temperature-dependent natural emissions, such as soil Nitrous acid (HONO) (Tan et al., 2023), may also lead to an underestimation of ozone responses to extreme temperatures in the GEOS-Chem simulations. Uncertainties in the biomass burning emission inventory (Fasullo et al., 2022) limit the accuracy of ozone-temperature sensitivity simulations in fire-impacted regions, such as the mountainous western United States. The 50 km resolution of the model may not fully capture sub-grid meteorological variations, which can play an important role in reproducing extreme conditions at site-level scales. Our study demonstrates that ozone-temperature sensitivity is highly responsive to changes in emissions, emphasizing the importance of more accurate anthropogenic emissions inventory for interpreting the ozone-temperature relationship. Further efforts are needed to enhance the model's ability to capture long-term trends in the ozone response to

**temperature (including underlying weather conditions and transport patterns), and to better unravel the mechanisms driving the observed ozone-temperature relationship, in particular the role of transport and ventilation."**

Reference:

Christiansen, A., Mickley, L. J., and Hu, L.: Constraining long-term NOx emissions over the United States and Europe using nitrate wet deposition monitoring networks, Atmospheric Chemistry and Physics, 24, 4569–4589, https://doi.org/10.5194/acp-24-4569-2024, 2024.

Clifton, O. E., Fiore, A. M., Massman, W. J., Baublitz, C. B., Coyle, M., Emberson, L., Fares, S., Farmer, D. K., Gentine, P., Gerosa, G., Guenther, A. B., Helmig, D., Lombardozzi, D. L., Munger, J. W., Patton, E. G., Pusede, S. E., Schwede, D. B., Silva, S. J., Sörgel, M., Steiner, A. L., and Tai, A. P. K.: Dry Deposition of Ozone Over Land: Processes, Measurement, and Modeling, Reviews of Geophysics, 58, e2019RG000670, https://doi.org/10.1029/2019RG000670, 2020.

Fasullo, J. T., Lamarque, J.-F., Hannay, C., Rosenbloom, N., Tilmes, S., DeRepentigny, P., Jahn, A., and Deser, C.: Spurious Late Historical-Era Warming in CESM2 Driven by Prescribed Biomass Burning Emissions, Geophysical Research Letters, 49, e2021GL097420, https://doi.org/10.1029/2021GL097420, 2022.

Kerr, G. H., Waugh, D. W., Strode, S. A., Steenrod, S. D., Oman, L. D., and Strahan, S. E.: Disentangling the Drivers of the Summertime Ozone-Temperature Relationship Over the United States, J. Geophys. Res. Atmos., 124, 10503–10524, https://doi.org/10.1029/2019JD030572, 2019.

Oikawa, P. Y., Ge, C., Wang, J., Eberwein, J. R., Liang, L. L., Allsman, L. A., Grantz, D. A., and Jenerette, G. D.: Unusually high soil nitrogen oxide emissions influence air quality in a high-temperature agricultural region, Nat Commun, 6, 8753, https://doi.org/10.1038/ncomms9753, 2015.

Tan, W., Wang, H., Su, J., Sun, R., He, C., Lu, X., Lin, J., Xue, C., Wang, H., Liu, Y., Liu, L., Zhang, L., Wu, D., Mu, Y., and Fan, S.: Soil Emissions of Reactive Nitrogen Accelerate Summertime Surface Ozone Increases in the North China Plain, Environ. Sci. Technol., 57, 12782–12793, https://doi.org/10.1021/acs.est.3c01823, 2023.

Wu, W., Fu, T.-M., Arnold, S. R., Spracklen, D. V., Zhang, A., Tao, W., Wang, X., Hou, Y., Mo, J., Chen, J., Li, Y., Feng, X., Lin, H., Huang, Z., Zheng, J., Shen, H., Zhu, L., Wang, C., Ye, J., and Yang, X.: Temperature-Dependent Evaporative Anthropogenic VOC Emissions Significantly Exacerbate Regional Ozone Pollution, Environ. Sci. Technol., https://doi.org/10.1021/acs.est.3c09122, 2024.

Wang, Y., Ge, C., Garcia, L. C., Jenerette, G. D., Oikawa, P. Y., and Wang, J.: Improved modelling of

soil NOx emissions in a high temperature agricultural region: role of background emissions on NO2

trend over the US, Environ. Res. Lett., 16, 084061, https://doi.org/10.1088/1748-9326/ac16a3, 2021.

**Comment [1-3]:** Line 155 suggests that the simulations have only been spun-up for one month; it's also unclear what initial condition is used (i.e. consistent with what year of meteorology and emissions). The manuscript needs to justify that the short spin-up time does not impact the results and that the simulation is at steady-state with the emissions. The authors cite the short lifetime for O3 in the boundary layer (and longer aloft) (lines 155-157). However, given that they do not parse how much of the surface O3 in the simulations is locally produced vs transported (regionally, intercontinentally, from the stratosphere), and that one of the important temperature-sensitive drivers is PAN, the 1 month simulation spin-up is not necessarily sufficient. The authors should test this for the year of maximum difference from the initial conditions (i.e. if the initial condition is consistent with 1990 emissions, then perform this sensitivity simulation for 2021): a global simulation that is spun-up for 6 months prior to generating the boundary conditions for the July simulation (to verify that changing transport of ozone and ozone precursors do not impact the results). These results should be included in the SI to justify the approach used here, and, in the unfortunate case that the results are impacted by the spin-up time, the authors would need to perform longer spin-ups for all their simulations.

**Response [1-3]: Thank you for pointing it out. We agree that increasing the model simulation spin-up time to 6 months or longer is more reasonable. However, our study includes 17 simulations with different configurations at a resolution of 0.5°(latitude) × 0.625°(longitude), with three simulations are conduced biennially in 1995-2017. Re-running all the simulations will be a significant challenge to time and computational sources. We follow your suggestion to validate the reliability of our simulation experiments with a relatively short spin-up time. For this purpose, we first performed an 8-month global simulation starting from January 2017, which provided boundary conditions for high-resolution simulations in June and July (so that the spin-up time for global simulation is 6-month). The initial conditions for the high-resolution (0.5°x0.625°) simulations were obtained by interpolating the spin-up of the 6-month global simulation onto the high-resolution grid. We compared the surface ozone concentrations and ozone-temperature sensitivity between this long**

spin-up time simulation and the BASE simulation with 1-month spin-up in Figure S2. The results show that the differences between the simulations with 1-month and 6-month spin-up times have only minor impacts on ozone concentrations and $m_{\Delta O3-\Delta Tmax}$. The average differences between the two simulations were only 0.3% for ozone concentrations and 2.3% for $m_{\Delta O3-\Delta Tmax}$, with extremely high spatial consistency (r > 0.99). This confirms that using a 1-month spin-up period does not affect our analysis and conclusions. Nevertheless, we acknowledge longer spin-up time should be taken in future modeling studies.

We have added the following content to the main text to demonstrate the reliability of our experiments in section 2.4: "To demonstrate this, we conducted an additional set of experiments, starting with a global simulation at 2°×2.5° resolution from 1st January 2017 to 1st August 2017. The global simulation on 1st June 2017 was then interpolated into the high-resolution nested grid to drive the high-resolution simulation from 1st June 2017 to 1st August 2017. A comparison of surface MDA8 ozone concentrations and ozone-temperature sensitivity between the two sets of simulations is shown in Figure S2. We find that the differences between the simulations with 1-month and 6-month spin-up times had only minor impacts on ozone concentrations and $m_{\Delta O3-\Delta Tmax}$. The average differences between the two simulations were only 2.3% for ozone concentrations and 0.3% for $m_{\Delta O3-\Delta Tmax}$, with high spatial consistency (r > 0.99). This confirms that using a 1-month spin-up time for the simulation should not affect the analysis and conclusions. However, a longer spin-up time is favorable for generating global chemical fields when sufficient computational resources are available."

[Figure]

Figure S2. The impacts of different spin-up time for MDA8 ozone and $m_{\Delta O3-\Delta Tmax}$. The difference

**between BASE and Long spin-up time simulations in (a) MDA8 ozone and (b) $m_{\Delta O3\text{-}\Delta Tmax}$. The correlation coefficients (r) between the simulations and mean values for the CONUS sites are shown inset.**

**Comment [1-4]:** 3. Figure 3a and associated text: Can the authors explain the dip and rebound in the O3-T relationship from 1996-2004? Is this trend present in all regions?

**Response [1-4]: We find significant spatiotemporal variability in ozone-temperature sensitivity trends across different regions. The overall declining trend in the CONUS ozone-temperature sensitivity slowed down or even reversed during 1996-2004, primarily due to an increase in ozone-temperature sensitivity across several regions, including the SEUS, Plains, and Midwest regions. Fu et al. (2015) pointed out that the increase in the SEUS region from 1990 to 2000 was mainly driven by changes in meteorological conditions, and this meteorological effect may have extended to other parts of the eastern United States during 1996-2004. In contrast, the western regions (the Intermountain West and SWUS) were not affected. We have added the following discussion in Section 3.1: "However, we notice an increase of $m_{\Delta O3\text{-}\Delta Tmax}$ in 1990-2000 for the SEUS region and in 1999-2005 for the Plains region (Figure S4). The increase in ozone-temperature sensitivity in these two regions explains the $m_{\Delta O3\text{-}\Delta Tmax}$ plateau in CONUS during the 1996-2004 period. Fu et al. (2015) attributes the increase ozone-temperature sensitivity in 1990-2000 in the SEUS to variations in regional ozone advection tied to climate variability. This further underscores the significant influence of climate variability on $m_{\Delta O3\text{-}\Delta Tmax}$ trends."**

**Reference:**

Fu, T.-M., Zheng, Y., Paulot, F., Mao, J., and Yantosca, R. M.: Positive but variable sensitivity of August surface ozone to large-scale warming in the southeast United States, Nature Clim Change, 5, 454–458, https://doi.org/10.1038/nclimate2567, 2015.

**Comment [1-5]:**

Line 94: "derive" seems inappropriate since the authors did not produce the MERRA2 product. I suggest "use" would be more accurate

Line 129: language "is capable of"

Line 151: "gas" should be plural

Line 180: language: replace "with both in" to "at both the"

Line 312: language: replace "propose by" with "theory using"

Line 314: language: "GEOS-Chem model simulates no"

Line 318: language: replace "neglectable" with "negligible"

**Response [1-5]: Thank you for pointing it out. We have corrected them accordingly.**

---

## Author Comment (AC2)

**Reviewer #2**

**Comment [2-1]:** This study explores the sensitivity of summertime ozone pollution in the United States to changes in temperature, focusing in particular on changes in that sensitivity across three recent decades. Manuscript text is generally clear and cohesive and accompanying figures are well constructed and easy to interpret. On the whole I find this an interesting and useful expansion of previous ozone-temperature studies and a worthwhile addition to the literature. I do have a few suggestions for strengthening the paper before publication:

**Response [2-1]: We thank the reviewer for the positive and valuable comments. All of them have been implemented in the revised manuscript. Please see our itemized responses below.**

**Comment [2-2]:** The CEDS inventory has some known biases in terms of agreement with observations. Of particular relevance for this study, previous work has found regional patterns in NOx biases, pointing overall to overestimates in the US (e.g. Christiansen et al., 2024 https://doi.org/10.5194/acp-24-4569-2024). Considering its importance to this study, it would be worth exploring previous work evaluating the CEDS inventory with respect to ozone precursors and commenting on how any biases or uncertainties may be influencing results shown here.

**Response [2-2]: Thank you pointing it out. The overestimation of anthropogenic emissions in the post-2010 emission inventories may be a key reason for the underestimation of the ozone-temperature sensitivity trends. We have added discussions on the uncertainties in anthropogenic NO$_x$ emissions from the CEDS inventory and their potential impacts on the ozone-temperature sensitivity.**

**In Section 3.2: "The simulated ozone-temperature sensitivity for 2013–2017 shows an overestimation, particularly in the SEUS and Midwest regions (Figure S8). Christiansen et al. (2024) suggested that the CEDS inventory overestimates post-2010 anthropogenic NO$_x$ emissions, especially in the eastern United States, which may lead to overestimation of ozone-temperature sensitivity in these regions."**

**In section 4: "Our study demonstrates that ozone-temperature sensitivity is highly responsive to**

**changes in emissions, emphasizing the importance of more accurate anthropogenic emissions inventory for interpreting the ozone-temperature relationship."**

**Reference:**

Christiansen, A., Mickley, L. J., and Hu, L.: Constraining long-term NOx emissions over the United States and Europe using nitrate wet deposition monitoring networks, Atmospheric Chemistry and Physics, 24, 4569–4589, https://doi.org/10.5194/acp-24-4569-2024, 2024.

**Comment [2-3]:** The naming scheme for normalized cases confused me somewhat. For most cases it appears to identify the effect being normalized or removed (FTEMP normalizes temperature fields), but for FTRANS this appears to be the opposite, as all meteorology is normalized except for transport. Some clarification and consistency here would help for parsing later results.

**Response [2-3]: Thank you for your suggestion. We have renamed the BASE-FTRANS and 1995E-FTRANS simulations to BASE-TRANS and 1995E-TRANS (all meteorology is normalized except for transport).**

**Comment [2-4]:** On a related note, it appears that a number of simulations listed in Table 1 are not explicitly mentioned or discussed in the manuscript text. If these simulations turned out to be used in developing manuscript figures and conclusions, it should be clearer how and where they were incorporated, with explicit case names cited for easier reference back to the table.

**Response [2-4]: Thank you for your suggestion. Our results are primarily presented by comparing the differences between various simulations, but the large number of simulations may cause some confusion for readers. To address this, we have added a summary of the differences between the simulations used for quantifying the drivers of $m_{\Delta O3-\Delta Tmax}$ trends in Table S2.**

**Table S2** The contribution for each mechanism

| Term | Calculation method |
| --- | --- |

| | |
|---|---|
| All effect contribution | the difference in $m_{\Delta O3\text{-}\Delta Tmax}$ between the BASE and 1995E simulation |
| Temperature-indirect effect contribution | the difference of $m_{\Delta O3\text{-}\Delta Tmax}$ between BASE-FTEMP and 1995E-FTEMP |
| Temperature-direct effect contribution | the difference of $m_{\Delta O3\text{-}\Delta Tmax}$ between BASE and 1995E minus the difference between BASE-FTEMP and 1995E-FTEMP |
| Transport contribution | the difference of $m_{\Delta O3\text{-}\Delta Tmax}$ between BASE-TRANS and 1995E-TRANS |
| Other-indirect effect contribution | difference of $m_{\Delta O3\text{-}\Delta Tmax}$ between BASE-FTEMP and 1995E-FTEMP minus the difference between BASE-TRANS and 1995E- TRANS |
| Combined contribution from four temperature-direct effects | the difference of $m_{\Delta O3\text{-}\Delta Tmax}$ between BASE and 1995E minus the difference between BASE-F4PATHS and 1995E-F4PATHS |
| BVOCs contribution | the difference of $m_{\Delta O3\text{-}\Delta Tmax}$ between BASE and 1995E minus the difference between BASE-FBVOC and 1995E-FBVOC |
| Soil $NO_x$ contribution | the difference of $m_{\Delta O3\text{-}\Delta Tmax}$ between BASE and 1995E minus the difference between BASE-FSNO$_x$ and 1995E-FSNO$_x$ |
| PAN decomposition contribution | the difference of $m_{\Delta O3\text{-}\Delta Tmax}$ between BASE and 1995E minus the difference between BASE-FPAN and 1995E- FPAN |
| Dry deposition contribution | the difference of $m_{\Delta O3\text{-}\Delta Tmax}$ between BASE and 1995E minus the difference between BASE-FDEP and 1995E- FDEP |

**Comment [2-5]:** While the details of transport effects are not a focal point of this paper, I found the

description of transport impacts (lines 365-376) to be a bit thin and muddled relative to other sections, especially considering their apparent importance. Do BASE-TRANS and 1995E-TRANS refer to BASE-FTRANS and 1995E-FTRANS from Table 1? Why would solar radiation and BVOC emissions in the SE be relevant to the patterns shown in 7a, since (if I understand these cases correctly) all meteorology other than transport has been normalized out in the simulations being subtracted here? A bit more attention to these results, identification of possible mechanisms at play, and discussion within the context of the broader literature would be appreciated.

**Response [2-5]: Thank you pointing it out. The impact of transport on ozone-temperature sensitivity largely depends on the transport patterns that has significant temporal variation. Discussing transport effects based on simulation over just one month (July 2017) may not provide sufficiently robust information. Thus, we have only provided a brief discussion. We apologize for the confusion regarding Figure 7, where the descriptions were incorrect: BASE-TRANS and 1995E-TRANS should refer to BASE-FTRANS and 1995E-FTRANS from Table 1. We have separated the indirect effects contributing to the reduction in ozone-temperature sensitivity due to anthropogenic emission reductions into transport and other indirect effects. The influence of solar radiation on BVOC emissions in the southeastern United States is related to the contribution from other indirect effects (Figure 7b, not Figure 7a). This is because the radiation received by vegetation is highly correlated with temperature, and radiation plays a crucial role in BVOC emission calculations in the model (Guenther et al., 2012). This strong collinearity likely explains the significant contribution of other indirect effects in the southeastern United States. We have added further discussion in the main text to highlight this point in Section 3.3: "The temperature-indirect effect excluding transport (Figure 7b) on $m_{\Delta O3-\Delta tmax}$ shows a more uniform decline with reduced emissions in most regions across the CONUS, with a larger decrease in Southeast US. The radiation received by vegetation in the southeastern United States is highly collinear with temperature and also plays an important role in BVOC emissions (Guenther et al., 2012), which may reflect its potential for ozone formation reduces with the decline in anthropogenic $NO_x$ emissions. In comparison, the transport effect has larger impacts on the $m_{\Delta O3-\Delta Tmax}$ trend (Figure 7a) with reduced $NO_x$ emissions in the northeastern US, where transport has the largest contribution to the mean $m_{\Delta O3-\Delta Tmax}$ values (Figure S10) as also reported in Kerr et al. (2019). Some studies have**

**demonstrated that changes in mid-latitude weather systems can significantly influence the ozone-temperature sensitivity by affecting pollutant transport (Barnes and Fiore, 2013; Kerr et al., 2020), which could be the underlying mechanism explaining the role of transport in contributing to the decrease of ozone-temperature sensitivity with emission reductions."**

**Reference:**

Guenther, A. B., Jiang, X., Heald, C. L., Sakulyanontvittaya, T., Duhl, T., Emmons, L. K., and Wang, X.: The Model of Emissions of Gases and Aerosols from Nature version 2.1 (MEGAN2.1): an extended and updated framework for modeling biogenic emissions, Geoscientific Model Development, 5, 1471–1492, https://doi.org/10.5194/gmd-5-1471-2012, 2012.

Barnes, E. A. and Polvani, L.: Response of the Midlatitude Jets, and of Their Variability, to Increased Greenhouse Gases in the CMIP5 Models, https://doi.org/10.1175/JCLI-D-12-00536.1, 2013.

Kerr, G. H., Waugh, D. W., Strode, S. A., Steenrod, S. D., Oman, L. D., and Strahan, S. E.: Disentangling the Drivers of the Summertime Ozone-Temperature Relationship Over the United States, J. Geophys. Res. Atmos., 124, 10503–10524, https://doi.org/10.1029/2019JD030572, 2019.

Kerr, G. H., Waugh, D. W., Steenrod, S. D., Strode, S. A., and Strahan, S. E.: Surface Ozone-Meteorology Relationships: Spatial Variations and the Role of the Jet Stream, Journal of Geophysical Research: Atmospheres, 125, e2020JD032735, https://doi.org/10.1029/2020JD032735, 2020.

---

## Author Comment (AC3)

**Reviewer #3**

**Comment [3-1]:** This manuscript re-visits earlier work of Porter and Heald (2019) and extends it to examine specific factors driving observed trends in local relationships between ozone and temperature. While the idea that ozone-temperature relationships are in part fueled by the availability of NOx is not new (Wu et al., 2008; Zanis et al., 2022), the advance here involves quantification of the impact of the known NOx reductions over recent decades in the USA to weakening the ozone-temperature relationship recorded at local monitoring sites. Quantifying the role of the selected individual 'direct' and a few 'indirect' processes as represented in the GEOS-Chem model to the changes in these relationships as shown in Figures 5, 6, 7 is a useful benchmark against which future work may gauge the importance of changes in these and other processes in the coming years as well as to compare to findings in other models.

**Response [3-1]: We thank the reviewer for the positive and valuable comments. All of them have been implemented in the revised manuscript. Please see our itemized responses below.**

**Comment [3-2]:** The mean bias of the temperature fields is evaluated (line 100; Figure S1) but aren't the trends in near-surface temperature over this period more relevant to the present study? As it is, the mean biases could lead to errors in the ozone simulation as noted by Rasmussen et al. (2012).

**Response [3-2]: We agree. We have added the following Table and analysis in Section 2.2 "We also compare temperature trends from MERRA-2 with observations over the period 1990-2021 (Table S1). While the overall trends are consistent, there are notable overestimation (e.g. NEUS, Plains) and underestimation (e.g. SEUS and SWUS) in different regions, which may lead to biases in interpreting the observed ozone-temperature sensitivity (as observed ozone variation responds to "true" air temperature)."**

**Table S1** Observed vs MERRA-2 $T_{max}$ (daily maximum temperature) trend at summertime (June, July, August) (K/decade) from 1990 to 2021 in different regions.

|         | CONUS  | NEUS | Midwest | SEUS | Plains | Intermountain West | NWUS | SWUS   |
|---------|--------|------|---------|------|--------|--------------------|------|--------|
| **OBS** | 0.20   | 0.01 | 0.68**  | 0.57 | -0.09  | 0.48**             | 0.43 | 1.44** |
| **MERRA-2** | 0.27* | 0.10 | 0.58* | 0.08 | 0.23 | 0.33            | 0.58 | 1.24** |

$^{**}$represents p-value<0.01, $^{*}$represents p-value<0.05

**Comment [3-3]:** The lateral boundary conditions used to drive the regional nested simulation should be described in a bit more detail. Was this a continuous run, or was the global model also run for 1-month spin up and June plus July every 2 years?

**Response [3-3]: Thank you for pointing it out. The lateral boundary conditions we used were simulated every two years, with each simulation covering June and July. The June simulation was treated as a spin-up period and thus discarded. Acknowledging the spin-up time of one month is relatively short, we conducted a separate simulation with a 6-month spin-up time and compared it with the BASE simulation. We find neglectable differences in ozone concentrations (0.3%) and $m_{AO3-\Delta Tmax}$ (2.3%) between the two simulations, suggesting that the spin-up time should not significantly affect our analysis and conclusions. We have revised the relevant description and added the following discussions in Section 2.4. "To demonstrate this, we conducted an additional set of experiments, starting with a global simulation at 2°×2.5° resolution from 1$^{st}$ January 2017 to 1$^{st}$ August 2017. The global simulation on 1$^{st}$ June 2017 was then interpolated into the high-resolution nested grid to drive the high-resolution simulation from 1$^{st}$ June 2017 to 1$^{st}$ August 2017. A comparison of surface MDA8 ozone concentrations and ozone-temperature sensitivity between the two sets of simulations is shown in Figure S2. We find that the differences between the simulations with 1-month and 6-month spin-up times had only minor impacts on ozone concentrations and $m_{AO3-\Delta Tmax}$. The average differences between the two simulations were only 2.3% for ozone concentrations and 0.3% for $m_{AO3-\Delta Tmax}$, with high spatial consistency (r > 0.99). This confirms that using a 1-month spin-up time for the simulation should not affect the analysis and conclusions. However, a longer spin-up time is favorable for generating global chemical fields when sufficient computational resources are available."**

[Figure]

**Figure S2. The impacts of different spin-up time for MDA8 ozone and $m_{\Delta O3\text{-}\Delta Tmax}$. The difference between BASE and Long spin-up time simulations in (a) MDA8 ozone and (b) $m_{\Delta O3\text{-}\Delta Tmax}$. The correlation coefficients (r) between the simulations and mean values for the CONUS sites are shown inset.**

**Comment [3-4]:** The authors have missed some prior literature investigating how specific regional conditions shape relationships between ozone and specific meteorological variables such as temperature. Camalier et al. (2007) pointed out the weaker ozone-temperature relationship in the Southeast, which Tawfik and Steiner (2013) linked to differences in the coupling between the atmosphere and land (specifically soil moisture regimes) and suggest that surface drying is a more important predictor. Furthermore, the strong ozone-temperature relationships in the northern part of the domain has been linked to dynamics associated with the mid-latitude jet (Barnes and Fiore, 2013) and meridional transport (Kerr et al., 2020; Zhang et al., 2022). Some discussion of how the findings of this study fit in the context of those papers would be useful.

**Response [3-4]: Thank you for providing the references. We have added a discussion of these studies in Section 3.1: "The higher $m_{\Delta O3\text{-}\Delta Tmax}$ in the NEUS and Midwest regions than in other regions may reflect the stronger daily variation of ozone due to rapid shift of synoptic patterns (e.g. mid-latitude cyclones) in this region during summer (Leibensperger et al., 2008). Additionally, changes in other mid-latitude dynamic systems, such as meridional movement by the mid-latitude**

**jet, also play a significant role in shaping the regional ozone-temperature sensitivity (Barnes and Fiore, 2013; Kerr et al., 2020; Zhang et al., 2022c). We observe a decreasing gradient in both $r_{\Delta O3\text{-}\Delta Tmax}$ and $m_{\Delta O3\text{-}\Delta Tmax}$ from north to south in the eastern United States, which aligns with previous findings (Camalier et al., 2007; Tawfik and Steiner, 2013). This observed north-to-south shift may be related to the transition in land-atmosphere coupling mechanisms due to soil moisture limitations in the southern regions (Tawfik and Steiner, 2013)."**

**Reference:**

Barnes, E. A. and Polvani, L.: Response of the Midlatitude Jets, and of Their Variability, to Increased Greenhouse Gases in the CMIP5 Models, https://doi.org/10.1175/JCLI-D-12-00536.1, 2013.

Camalier, L., Cox, W., and Dolwick, P.: The effects of meteorology on ozone in urban areas and their use in assessing ozone trends, Atmospheric Environment, 41, 7127–7137, https://doi.org/10.1016/j.atmosenv.2007.04.061, 2007.

Leibensperger, E. M., Mickley, L. J., and Jacob, D. J.: Sensitivity of US air quality to mid-latitude cyclone frequency and implications of 1980–2006 climate change, Atmospheric Chemistry and Physics, 8, 7075–7086, https://doi.org/10.5194/acp-8-7075-2008, 2008.

Kerr, G. H., Waugh, D. W., Steenrod, S. D., Strode, S. A., and Strahan, S. E.: Surface Ozone-Meteorology Relationships: Spatial Variations and the Role of the Jet Stream, Journal of Geophysical Research: Atmospheres, 125, e2020JD032735, https://doi.org/10.1029/2020JD032735, 2020.

Tawfik, A. B. and Steiner, A. L.: A proposed physical mechanism for ozone-meteorology correlations using land–atmosphere coupling regimes, Atmospheric Environment, 72, 50–59, https://doi.org/10.1016/j.atmosenv.2013.03.002, 2013.

Zhang, X., Waugh, D. W., Kerr, G. H., and Miller, S. M.: Surface Ozone-Temperature Relationship: The Meridional Gradient Ratio Approximation, Geophysical Research Letters, 49, e2022GL098680, https://doi.org/10.1029/2022GL098680, 2022.

**Comment [3-5]:** Uncertainties in the model and their implications for the conclusions could be discussed more clearly.  For example, the underlying assumption is that the model represents all important processes driving ozone-temperature relationships. The BB4CMIP emissions have spurious variations associated with the introduction of GFED emissions (satellite data) after 1997 (Fasullo et al., 2022),

which might lead to problems for ozone trends in regions strongly influenced by fire. The dry deposition scheme only includes stomatal deposition variations with with meteorology (line 150) but non-stomatal pathways may also respond to temperature (Clifton et al., 2022).

**Response [3-5]: Thank you for pointing out this issue. We have added a discussion on the model uncertainties in Section 4: "Nevertheless, there is significant room for improving the ability in capturing the ozone-temperature relationship in the chemical transport model. The GEOS-Chem simulations do not account for the response of anthropogenic $NO_x$ and VOCs emissions to temperature. Recent studies have shown that these emissions can increase simulated regional ozone-temperature sensitivity by up to 7% and 14% (Kerr et al., 2019; Wu et al., 2024). The parameterization of several temperature-dependent processes is limited or even missing in the model. For example, the dry deposition scheme used in this study lacks the temperature response of non-stomatal pathways (Clifton et al., 2020), which could introduce uncertainty in simulated $m_{\Delta O3-\Delta Tmax}$ particularly in vegetation-rich regions such as the southeastern United States. Additionally, according to the BDSNP scheme used in this study, soil $NO_x$ emissions are modeled as an exponential function of temperature between 0 and 30 °C, remaining constant at temperatures above 30 °C. However, some studies have reported continuous increases in soil $NO_x$ emissions at temperatures higher than 30 °C in regions such as California (Oikawa et al., 2015; Wang et al., 2021). The absence of other temperature-dependent natural emissions, such as soil Nitrous acid (HONO) (Tan et al., 2023), may also lead to an underestimation of ozone responses to extreme temperatures in the GEOS-Chem simulations. Uncertainties in the biomass burning emission inventory (Fasullo et al., 2022) limit the accuracy of ozone-temperature sensitivity simulations in fire-impacted regions, such as the mountainous western United States. The 50 km resolution of the model may not fully capture sub-grid meteorological variations, which can play an important role in reproducing extreme conditions at site-level scales. Our study demonstrates that ozone-temperature sensitivity is highly responsive to changes in emissions, emphasizing the importance of more accurate anthropogenic emissions inventory for interpreting the ozone-temperature relationship. Further efforts are needed to enhance the model's ability to capture long-term trends in the ozone response to temperature (including underlying weather conditions and transport patterns), and to better unravel the mechanisms driving the observed ozone-**

**temperature relationship, in particular the role of transport and ventilation."**

**Reference:**

Clifton, O. E., Fiore, A. M., Massman, W. J., Baublitz, C. B., Coyle, M., Emberson, L., Fares, S., Farmer, D. K., Gentine, P., Gerosa, G., Guenther, A. B., Helmig, D., Lombardozzi, D. L., Munger, J. W., Patton, E. G., Pusede, S. E., Schwede, D. B., Silva, S. J., Sörgel, M., Steiner, A. L., and Tai, A. P. K.: Dry Deposition of Ozone Over Land: Processes, Measurement, and Modeling, Reviews of Geophysics, 58, e2019RG000670, https://doi.org/10.1029/2019RG000670, 2020.

Fasullo, J. T., Lamarque, J.-F., Hannay, C., Rosenbloom, N., Tilmes, S., DeRepentigny, P., Jahn, A., and Deser, C.: Spurious Late Historical-Era Warming in CESM2 Driven by Prescribed Biomass Burning Emissions, Geophysical Research Letters, 49, e2021GL097420, https://doi.org/10.1029/2021GL097420, 2022.

Kerr, G. H., Waugh, D. W., Strode, S. A., Steenrod, S. D., Oman, L. D., and Strahan, S. E.: Disentangling the Drivers of the Summertime Ozone-Temperature Relationship Over the United States, J. Geophys. Res. Atmos., 124, 10503–10524, https://doi.org/10.1029/2019JD030572, 2019.

Oikawa, P. Y., Ge, C., Wang, J., Eberwein, J. R., Liang, L. L., Allsman, L. A., Grantz, D. A., and Jenerette, G. D.: Unusually high soil nitrogen oxide emissions influence air quality in a high-temperature agricultural region, Nat Commun, 6, 8753, https://doi.org/10.1038/ncomms9753, 2015.

Tan, W., Wang, H., Su, J., Sun, R., He, C., Lu, X., Lin, J., Xue, C., Wang, H., Liu, Y., Liu, L., Zhang, L., Wu, D., Mu, Y., and Fan, S.: Soil Emissions of Reactive Nitrogen Accelerate Summertime Surface Ozone Increases in the North China Plain, Environ. Sci. Technol., 57, 12782–12793, https://doi.org/10.1021/acs.est.3c01823, 2023.

Wu, W., Fu, T.-M., Arnold, S. R., Spracklen, D. V., Zhang, A., Tao, W., Wang, X., Hou, Y., Mo, J., Chen, J., Li, Y., Feng, X., Lin, H., Huang, Z., Zheng, J., Shen, H., Zhu, L., Wang, C., Ye, J., and Yang, X.: Temperature-Dependent Evaporative Anthropogenic VOC Emissions Significantly Exacerbate Regional Ozone Pollution, Environ. Sci. Technol., https://doi.org/10.1021/acs.est.3c09122, 2024.

Wang, Y., Ge, C., Garcia, L. C., Jenerette, G. D., Oikawa, P. Y., and Wang, J.: Improved modelling of soil NOx emissions in a high temperature agricultural region: role of background emissions on NO2 trend over the US, Environ. Res. Lett., 16, 084061, https://doi.org/10.1088/1748-9326/ac16a3, 2021.

**Comment [3-6]:** Why does the model miss the observed decline in the slope after 2010 in Figure 5b? Figure S7 suggests this is occurring in the SEUS and Midwest; some of the literature referenced may be helpful for additional context in interpreting the differences across regions from the perspective of the processes that dominate in different regions.

**Response [3-6]: We find that overestimation of $m_{\Delta O3-\Delta Tmax}$ after 2010 is likely due to the uncertainty in anthropogenic NO$_x$ emissions. We have added the following discussion in Section 3.2: "The simulated ozone-temperature sensitivity for 2013–2017 shows an overestimation, particularly in the SEUS and Midwest regions (Figure S8). Christiansen et al. (2024) suggested that the CEDS inventory overestimates post-2010 anthropogenic NO$_x$ emissions, especially in the eastern United States, which may lead to overestimation of ozone-temperature sensitivity in these regions. The GEOS-Chem model also misses several pathways in describing the responses of ozone to temperature, such as the responses in anthropogenic emission and land-atmosphere interaction through soil and vegetation. This will be discussed in detail in Section 4."**

**Reference:**

Christiansen, A., Mickley, L. J., and Hu, L.: Constraining long-term NOx emissions over the United States and Europe using nitrate wet deposition monitoring networks, Atmospheric Chemistry and Physics, 24, 4569–4589, https://doi.org/10.5194/acp-24-4569-2024, 2024.

**Comment [3-7]:** The focus on NYS in Section 3.4, while interesting, appears arbitrary. What is the rationale for choosing this state? Are the correlations between ozone and temperature particularly strong there?

**Response [3-7]: In Section 3.4, we focus on the impact of ozone pollution on human health under high-temperature conditions. We selected New York State (NYS) as an example because of its strong ozone-temperature correlations and high population density. This allows us to explore how emission reductions have led to a significant decrease in ozone concentrations during high-temperature events due to in the declined ozone-temperature sensitivity. We also found that this phenomenon is widespread across other regions (Figure 9b).**

**Comment [3-8]:** The goal of Figure 9 is very interesting, but additional work would help strengthen the analysis. What are the trends in the 0-10% temperature bin values as compared to the 90-100% bins? These are likely sampling very different meteorological conditions. How does the metric used in this figure compare to the linear fit between daily ozone and temperature?

**Response [3-8]: In the discussion of ozone mitigation benefit in Section 3.4, we compared the ozone-temperature relationship under different emission scenarios for the 2013-2017 climate conditions, with both sets of simulations using the same meteorological conditions (so no trends in temperature is involved). As you mentioned the 0-10% and 90-100% temperature bins represent significantly different meteorological conditions across regions (Figure R1a). We compared the ozone mitigation benefit, as defined in our study, with the reduction in ozone-temperature sensitivity calculated using the decreased $m_{\Delta O3-\Delta Tmax}$, and found that the two metrics are nearly identical (Figure R1b). We further analyzed the probability of ozone exceedance under high-temperature conditions before and after emission reductions, emphasizing the impact of the benefit by reducing $m_{\Delta O3-\Delta Tmax}$ in section 3.4: "This benefit significantly reduces the probability of ozone exceedance (MDA8 ozone > 70 ppbv) during high-temperature conditions (above the 90th percentile of $T_{max}$), from 70% (estimated from the 1995E simulation) to 28% (from the BASE simulation)."**

[Figure]

**Figure R1. (a) The distribution of the difference between the 0-10% and 90-100% temperatures bins. (b) Distributions of ozone mitigation benefit in July due to the decreased $m_{\Delta O3-\Delta Tmax}$, estimated as the difference $m_{\Delta O3-\Delta Tmax}$ between the 1995E and BASE multiplied by the temperature difference from the 0-10% to 90-100% bins at each grid in July (2013,2015 and 2017). Mean, max, and min**

**values for the 608 sites are shown inset.**

**Comment [3-9]:** The data availability statement regarding the model simulations, which are critical to the conclusions drawn in the paper, does not appear to align with current best practices in sharing data for open science. Will the authors provide at least the datasets behind their figures, or a limited set of diagnostics from their simulations in a public repository to allow future studies to easily re-visit and extend their findings?

**Response [3-9]: We agree. Upon acceptance, we will update the Data Availability section accordingly to include a link to the repository and ensure that all relevant data is accessible for further research.**

**Comment [3-10]:** Line 12: In what applications are ozone-temperature relationships being used to predict the impacts of future climate change? The overall weak correlations (Figure 2; Table S1; low r values indicate that even at best less than half the variance is captured) suggest this is not a very useful metric for prediction.

**Response [3-10]: Thank you for your comment. High ozone-temperature sensitivity often indicates a region's heightened risk of climate penalties. While this metric may not precisely predict the exact increase in ozone due to climate change, it still serves as a useful reference for assessing potential risks in future scenarios. To more accurately convey our point, we have replaced "predict" with "infer."**

**Comment [3-11]:** line 67: It seems relevant to compare the r-values for model versus observations too.

**Response [3-11]: We agree. We have revised as: "In this study, we analyze the present-day (2017-2021) and long-term trends (1990-2021) in the summertime surface ozone-temperature relationship in the continental US."**

**Comment [3-12]:** Lines 114-115: What type of linear regression method is used to quantify the trend?

**Response [3-12]: We used a univariate linear regression based on the least squares method to quantify the long-term trend in ozone-temperature sensitivity.**

**Comment [3-13]:** Lines 137-142. The discussion of BDSNP is very confusing. The scheme is described but then line 141 suggests it isn't used, "but here we do not implement this scheme...". Please explain more clearly.

**Response [3-13]: We apologize for the confusion. Our intention was to highlight that the current parameterization scheme for soil $NO_x$ has certain limitations, which may introduce additional uncertainties into the study's results. We have moved this discussion to Section 4, where we provide a more comprehensive overview of the potential model uncertainties.**

**Comment [3-14]:** Figure 2. The color bar hides the relatively weak correlations across much of the country.

**Response [3-14]: The color bar of $r_{\Delta O3-\Delta Tmax}$ ranges from -0.2 to 1.0, so light colors indicate weak correlations. We find that 568 of 608 sites are with p-value<0.1, as indicated by the borders.**

**Comment [3-15]:** Figures 3b & 4. Are the values plotted meaningful in regions where the correlations are weak? It may be worth considering a screening that only plots for p-values above some threshold (0.10?).

**Response [3-15]: Following your suggestion, we have revised the Figure 3b to include only sites with $r_{\Delta O3-\Delta Tmax}$ p-values<0.01.**

**Comment [3-16]:** In Figure 4, the errors on the values of the slopes seem fairly large for the individual months (a lot of scatter).

**Response [3-16]: We agree. The scatter in the slope values for individual months is due to the**

model's weaker ability to capture the ozone-temperature response relationship in certain regions. Our intention is to provide an objective representation of the model's performance. We hope that future developments in atmospheric chemistry transport models will help reduce the bias.

**Comment [3-17]:** Lines 423-236. The increasing role of soil NOx on U.S. air quality has been noted in some other recent work as well; see for example Guo et al. (2018) and Geddes et al. (2022). Guo et al. (2018) also suggest that soil NOx may be contributing to ozone biases in GEOS-Chem.

**Response [3-17]: Thank you for pointing this out. We have cited these references in Section 3.3.**

**Comment [3-18]:** In Section 4, it would be useful to summarize how NOx has declined over this period, and whether the largest drops in the slopes/correlations have occurred in locations where anthropogenic NOx has decreased the most.

**Response [3-18]: Thank you for pointing this out. We have added a description in the main text: "During the period from 1990 to 2021, anthropogenic NO$_x$ emissions in the United States decreased by approximately 69%, and the eastern United States, where stricter anthropogenic emission controls were implemented, is the core region where ozone-temperature sensitivity has declined the most."**

**Comment [3-19]:**

Line 132 and elsewhere: biologic à biogenic?

Line 191 caption of Figure 2 black boarder à border

Line 255 least à smallest or weakest

Line 278 transportation à transport

Line 384 caption of Figure 8 temperature-indirect à temperature-direct ?

**Response [3-19]: Thank you for pointing it out. We corrected them accordingly.**

**Comment [3-20]:** Line 442 what is "ozone migration"?

**Response [3-20]: We have corrected it.**

**Comment [3-21]:** Line 468. Is this a spatial correlation of the slopes from the model vs observations?

**Response [3-21]: This is the ratio of the ozone-temperature sensitivity trends between model (-0.28 ppbv/K/decade) and the observations (-0.67 ppbv/K/decade).**

---

## Author Response (AR2)

Dear Editor Dr. Benjamin A Nault,

Thank you very much for handling our manuscript. Below, we provide our detailed responses to the reviewers' comments along with the revised manuscript, which includes all the changes made in response to the reviewers' suggestions. We have addressed every comment raised and incorporated the necessary revisions.

Thank you for your consideration.

Sincerely,

Shuai Li et al.
* * *
Report #2

Comment [2-1]: The authors have addressed the reviewer comments appropriately. I have minor comments to further improve the clarity of the paper.

Response [2-1]: We thank the reviewer for the positive and valuable comments. All of them have been implemented in the revised manuscript. Please see our itemized responses below.

Comment [2-2]: L46: change air stagnancy to air stagnation and remove enhanced before ventilation since the sentence refers to phenomena rather than the direction of change.

Response [2-2]: Thank you for pointing it out. We have corrected them accordingly.

Comment [2-3]: L68: it would be helpful to be consistent throughout the manuscript on the terminology: relationship versus sensitivity

Response [2-3]: Thank you for pointing it out. In this study, we aim to use the ozone-temperature

relationship to represent the overall association between ozone and temperature, including their correlation ($r_{\Delta O3-\Delta Tmax}$) and ozone-temperature sensitivity ($m_{\Delta O3-\Delta Tmax}$). In contrast, ozone-temperature sensitivity specifically refers to $m_{\Delta O3-\Delta Tmax}$. We have carefully reviewed and corrected the use of "relationship" and "sensitivity" throughout the manuscript to avoid any potential misunderstandings.

**Comment [2-4]:** L91: please clarify if AQS dataset provides "surface" temperature measurements.

**Response [2-4]: We change the description: "The AQS dataset also provides surface temperature measurements that could be ideally used in quantifying the ozone-temperature relationship at individual sites."**

**Comment [2-5]:** L157-159: NOx may have low solubility but wet deposition of its oxidation products (e.g., HNO3, PAN) have implications for ozone.

**Response [2-5]: We agree. Accordingly, we have revised the relevant description in the manuscript as follows: "NO$_x$ and ozone have low solubility, but wet deposition of NO$_x$ oxidation products may further influence ozone. We do not separately consider temperature's indirect influences on ozone through wet deposition processes in this study."**

**Comment [2-6]:** L160-161: The trend analysis period is 1990-2021 while the simualtions are done for 1995-2017. Please provide a specific reason or the shorter time priod for model simulations.

**Response [2-6]: Thank you for pointing this out. We have added the following text in Line 157 "We do not extend the simulation to earlier or later years due to lack of reliable anthropogenic emission inventory by the time when this study was designed."**

**Comment [2-7]:** L162: replace closed with close

**Response [2-7]: We have corrected them accordingly.**

**Comment [2-8]:** L179: "others" communicates vagueness. Please be specific.

**Response [2-8]: We change the relevant text in the manuscript: "including natural emissions of BVOCs and soil NOx, the chemical kinetics, dry deposition, and other mechanisms that may have minimal impacts on ozone."**

**Comment [2-9]:** L198: 1995E-AVOCs should be 1995EAVOCs. Figure 3b: Spatial is misspelt in the title near the top.

**Response [2-9]: We have corrected them accordingly.**

**Comment [2-10]:** Figure 5: How are the NOx emissions reported - kg NO or kg NO2?

**Response [2-10]: Thank you for pointing this out. We use kg NO to describe NOx emissions, and to avoid any confusion, we have changed the unit of NOx emissions to kg N in Figure 5.**

**Comment [2-11]:** L354: which version of CEDS? Is it the same used in this study?

**Response [2-11]: To ensure consistency in the study, all anthropogenic emission inventories used in this study are based on CEDS v2021_04_21 (O'Rourke, 2021).**

**Reference:**

O'Rourke, P. R., Smith, S. J., Mott, A., Ahsan, H., McDuffie, E. E., Crippa, M., Klimont, Z., McDonald, B., Wang, S., Nicholson, M. B., Feng, L., & Hoesly, R. M. (2021). CEDS v_2021_04_21 Release Emission Data (v_2021_02_05) [Data set]. Zenodo. https://doi.org/10.5281/zenodo.4741285

**Report #3**

**Comment [3-1]:** The authors have provided reasonable responses to all the questions and comments raised during review and I find the manuscript suitable for publication.

**Response [3-1]: We thank the reviewer for the positive and valuable comments. All of them have been implemented in the revised manuscript. Please see our itemized responses below.**

**Comment [3-2]:** I would request two further small clarifications on the description of the spin-ups.

1. The text is not perfectly clear about the initial condition for all the simulations that start on June 1. Does the text on lines 156-157 mean that the IC for every June 1 simulation (for every year) is July 1 of 2005? If so, could the authors please specify that the same initial condition is used for every simulation?

**Response [3-2]: In this study, all simulations use the same initial conditions. To clarify this, we have added the following description in Section 2.4 of the manuscript: "The initial chemical fields are close to conditions for July 2005 (the same initial fields used for each set of sensitivity experiments)"**

**Comment [3-3]:** 2. Line 165: while the average difference was small, Figure S2 shows some regional increases and decreases of ~+/-1 ppb in O3 and +/-0.3 ppb/K in O3/T. These are modest, but not negligible differences and the authors should clarify that the 2.3% average difference includes more significant regional increases and decreases.

**Response [3-3]: Thank you for pointing it out. We agree that a shorter spin-up time increases uncertainty for specific regions. To clarify this, we have added the following description in Section 2.4 of the manuscript:: "The average differences between the two simulations were only 0.3% for ozone concentrations and 2.3% for $m_{\Delta O3-\Delta Tmax}$, with high spatial consistency (r > 0.99). This confirms that using a 1-month spin-up time for the simulation should not affect the analysis and conclusions. However, for specific regions, more noticeable differences in ozone concentrations and $m_{\Delta O3-\Delta Tmax}$ exist between the two simulations. A longer spin-up time is favorable for generating global chemical fields when sufficient computational resources are available."**